



**Hydro-stochastic interpolation coupling with Budyko approach for spatial**
**prediction of mean annual runoff**
Ning Qiu[a,b], Xi Chen[a,b]*, Qi Hu[c], Jintao Liu[a,b], Richao Huang[a,b], Man Gao[a,b]
[a] *State Key Laboratory of Hydrology-Water Resources and Hydraulic Engineering*
*Hohai University, Nanjing 210098, China*
[b] *College of Hydrology and Water Resources, Hohai University, Nanjing 210098, China*
[c] *School of Natural Resources, University of Nebraska-Lincoln, Lincoln NE 68583, U.S.*
*\*Corresponding author      E-mail: xichen@hhu.edu.cn*



**Abstract**
Hydro-stochastic interpolation method based on traditional block-kriging has often
been used to predict mean annual runoff in river basins. A caveat in this method is that
the statistic technique provides little physical insight on relationships of the external
forcing of climate and landscape and basin runoff. In this study, the spatial runoff is
decomposed into a deterministic trend and stochastic fluctuations around it. The former
is described by the Budyko method (Fu's equation) and the latter by hydro-stochastic
interpolation. The coupled method of stochastic interpolation and the Budyko method is
applied to interpolate spatial runoff in the Huaihe River basin of China, based on outlet
streamflow and climate data at 40 sub-basins. Results show that the coupled method
significantly improves spatial interpolation accuracy of mean annual runoff. The
prediction errors from the coupled method are much smaller than that from the respective
predictions by the Budyko scheme and the hydro-stochastic interpolation. The cross-
validation outcome of the determination efficient, $R_{cv}^2$, from the coupled method is 0.93,
much larger than 0.81 and 0.54 from the Budyko method and the hydro-stochastic
interpolation, respectively. The prediction from the coupled method describes accurately
the runoff distribution in the Huaihe River basin. In comparison, predictions from the
Budyko method and from the hydro-stochastic interpolation show substantial
overestimate of low runoff and underestimate of high runoff. These comparison results
support that the coupled hydro-stochastic interpolation with the Budyko method offers
an effective and accurate way in spatial interpolation of mean annual runoff.
**Keywords:** coupled Budyko-Hydro-stochastic interpolation; Mean annual runoff;





prediction accuracy, regional runoff
**1. Introduction**

Runoff at the outlet of a basin is a crucial element measuring the hydrological cycle

in the basin. Estimating runoffs and associated distribution pattern of water resources in
ungauged basins has been one of the key problems in hydrology (Sivapalan et al., 2003)
and a thorny issue in water management and planning (Imbach, 2010; Greenwood et al.,
2011). Among the existing methods for such estimation and prediction of water resources
availability is the regional or global runoff mapping by spatial interpolation.

Geostatistical approaches are mostly used in spatial interpolation. It estimates the

value at a given location as a weighted sum of data values at surrounding locations. The
spatial interpolation, assuming similarity of a generalized stochastic field (Jones, 2009),
uses secondary information often referred to as "multivariate" (Li and Heap, 2008). The
variable of interest is represented as a random field of values. Spatial similarity is
measured by the variance between pairs of points as a function of their Euclidian distance
(such as in Ordinary Kriging). Kriging has been the popular linear unbiased estimator,
i.e. an interpolation method in which the expected bias is zero and the expected kriging
error is minimized (Skøien et al., 2006). Ordinary Kriging (OK) estimates the local
constant mean and corresponding residuals which are regarded as random. Since the
spatial mean could also tell the trend tendency or nonstationary variation in space,
Kriging methods have been further developed into various geostatistical interpolators,
such as Kriging with a trend by incorporating the local trend within the neighborhood
search window as a smoothly varying function of the coordinates. Block Kriging (BK)





is also suggested as an extension of OK for estimating a block value instead of a point
value by replacing the point-to-point covariance with the point-to-block covariance
(Wackernagel, 1995). Recently, the kriging with an external drift (KED) is introduced to
incorporate the local trend within the neighborhood search window as a linear function
of a smoothly varying secondary variable instead of as a function of the spatial
coordinates (Goovaerts, 1997; Li and Heap, 2008).

Since streamflow discharge observed at the outlet represents the comprehensive

outcome of both precipitation and land surface over a certain drainage basin, the
nonstationary trend from spatial heterogeneity, regarded here as a "deterministic term"
at locations, exists due to distinguishingly spatial variability in climate-landscape factors,
such as higher or lower runoff corresponding to larger or smaller rainfall over space. The
spatially nonstationary trend of runoff can be interpreted by hydrological regionalization
in terms of hydro-climate and landscape data at various basins, e.g., developing
empirical relationships between runoff and its controlling factors of climate, land use
and topography (Qiao 1982; Arnell, 1992). Those empirical relationships obtain a
consensus on the form of regression equation adopted. As a simple semi-empirical
approach, the Budyko theory that partitions precipitation into evapotranspiration ($E$) and
runoff in regional scales based on surface water and energy balance, has been frequently
used (Milly, 1994; Koster and Suarez, 1999; Zhang et al., 2001; Donohue et al., 2007;
Li et al., 2013; Greve et al., 2014). Because Budyko method describes hydro-climate
relationship over a large area, its use in prediction of runoff and $E$ in any specific
basins/areas still comes with large errors (Potter and Zhang, 2009; Jiang et al., 2015).



Many efforts have been devoted to improving Budyko method in prediction of regional
runoff. Improvements have been made by including land use/landcover, geomorphology
and climate variability in deriving parameters in Budyko method (Han et al., 2011; Li et
al., 2013; Yang et al., 2007; Han et al., 2011).

Unlike interpolating a field, such as precipitation and evaporation, to find its values

at "points" in space (Lenton and RodriguezIturbe, 1977; Creutin and Obled, 1982; Tabios
and Salas, 1985; Dingman et al, 1988; Barancourt et al, 1992 and Bloschl, 2005), spatial
interpolation of runoff, which is an integrated spatial continuous process in basins
consisting of nested sub-basins, must take into account of the river network structure
that constraints water balance between upstream and downstream. Previous studies have
indicated that without adequate spatial variation information of runoff, e.g., neglecting
the lateral streamflow aspects or processing basin runoff behavior as "points" in space
(Villeneuve et al, 1979, Hisdal and Tveito, 1993), the runoff interpolation may
overestimate the actual runoff (Arnell, 1995). In hydro-stochastic interpolation of runoff,
the upstream and downstream basin area has been treated differently from neighboring
basins (Sauquet et al., 2000). It has been shown that the Euclidian distances used in
conventional stochastic methods fail to measure the spatial distance of pairs of the
connected (sub)-basins in most cases.

Given the obvious nested structure of basins, Gottschalk (1993a, b) developed a

hydro-stochastic approach for runoff interpolation. It takes full account of the concept
that runoff is an integrated course in the hierarchical structure of the river basin system.
Distance between a pair of basins is measured along the river network by sub-basin





distance instead of Euclidian distance. A drainage basin covariogram is replacing the
covariogram among "points" in conventional spatial interpolation. However, in this
concept, spatial runoff was considered as the spatial homogeneity (Sauquet, 2006) or
stationarity in the "deterministic term" of the regional average runoff over basins.
The inclusion of the deterministic term in the original geostatistical models has been
shown to increase interpolation accuracy of the basin variables, e.g., mean annual runoff
(Sauquet, 2006) and stream temperature (Laaha et al., 2013). Nevertheless, the
deterministic term is mostly described by an empirical formula linking the spatial
features, e.g. variability of mean annual runoff with elevation (Sauquet, 2006) and
relationship between the mean annual stream temperature and altitude of the gauge
(Laaha et al., 2013). The aims of this study are to incorporate the stochastic interpolation
method with semi-empirical approach in quantifying the deterministic trend for spatial
interpolation of runoff in Huaihe River, China. In this study, the spatial runoff at sub-
basins is separated into the deterministic trend and its residuals, which are estimated by
the Budyko framework and the errors between the observed runoff and the Budyko
estimation. The residuals or errors were used in the hydro-stochastic interpolation
proposed by Gottschalk (1993a, b, 2000). After that, the runoff prediction of any specific
sub-basins is calculated as the total of the interpolated residual and the Budyko
estimation. The improved method is tested in the Huaihe River basin, China. For
comparison, the leave-one-out cross validation approach was applied to evaluate
performance of the three interpolation methods: Budyko equation, hydro-stochastic
interpolation, and hydro-stochastics coupling with the Budyko equation.





## 2. Methodology

### 2.1 Spatial estimation of mean annual runoff by Budyko approach

Budyko approach explains the variability of mean annual water balance on terrestrial scale. It describes dependence of actual evapotranspiration ($E$) on precipitation ($P$) and potential evapotranspiration ($E_0$) (Williams et al., 2012). The original relationship ($E/P \sim E_0/P$) derived by Budyko (1974) is deterministic and nonparametric. After the Budyko curve was applied in various basins and climate conditions, it was found also dependent on local conditions, e.g., land use/cover (Donohue et al., 2007; Li et al., 2013), soil properties (Porporato et al., 2004; Donohue et al., 2012), topography (Shao et al., 2012; Xu et al., 2013), hydro-climatic variations of seasonality (Milly, 1994; Gentine et al., 2012; Berghuijs et al., 2014) and groundwater levels (Istanbulluoglu et al., 2012). Consequently, the Budyko curve has been extended to include those effects. Some of such effects were included in parametric forms (Fu, 1981; Choudhury, 1999; Yang et al., 2008; Gerrits et al., 2009; Wang and Tang, 2014). Among all revised Budyko curves, the one-parameter equation derived by Fu (Fu, 1981, Zhang et al. 2004) has been popularly used. This equation is as follows:

$$\frac{E}{P} = 1 + \frac{E_0}{P} - \left(1 + \left(\frac{E_0}{P}\right)^\omega\right)^{\frac{1}{\omega}} \tag{1}$$

or

$$R = \left(1 + \left(\frac{E_0}{P}\right)^\omega\right)^{\frac{1}{\omega}} - E_0 \tag{2}$$

where $P$, $E$, $E_0$ and $R$ are mean annual precipitation, actual evapotranspiration, potential evapotranspiration, and runoff (unit: mm), respectively, and $\omega$ is a dimensionless model parameter within the range $(1, \infty)$. In this formula, the





larger the ω is, the less the partitioning of precipitation into runoff.
The parameter ω can be calibrated by observed runoff at gauged basins or sub-
basins in the study area. With known ω, Eq. (2) can be used for prediction of ungauged
basin runoff or interpolation of spatial variation of runoff by using meteorological data
in the target sub basins (Parajka and Szolgay, 1998).
**2.2 Hydro-stochastic interpolation method**
Gottschalk (1993a) proposed the hydro-stochastic interpolation method for spatial
prediction of runoff based on Kriging interpolation. The Gottschalk's interpolation
method redefined a relevant distance between drainage basins to identify the river system
structure and supplement water balance constraints as follows.
As a spatial integrated continuous process, the predicted runoff in the specific unit
$r^*(A_0)$ in a basin can be expressed as

$$r^*(A_0) = \sum_{i=1}^{n} \lambda_i r(A_i) = \Lambda^T R \qquad (3)$$

where $A_0$ is the area of the specific unit, e.g. the basin area in this study, $r^*(A_0)$ is the
predicted runoff from that basin area, $r(A_i)$ is the observed runoff in a gauged basin $i$
with an area $A_i$ ($i = 1, \dots n$, $n$ is the total number of gauged basins), $\lambda_i$ is the weights
of a gauged basin $i$, and $\Lambda$ is the transposed column vector of the weights and $R$ is
the column vector of runoff $r(A_i)$.
Since $r^*(A_0)$ is the estimator of the true value $r(A_0)$, the best linear unbiased
estimator requires: $E[r^*(A_0) - r(A_0)] = 0$. To achieve the goal of minimizing the
estimate error, the following set of equations was developed to solve for the optimal
weights given that hydrologic variables satisfy the second order stationary assumption





(Ripley, 1976)
$$\begin{cases} \sum_{j=1}^{n} \lambda_i C(u_i, u_j) - \mu = C_0(u_i, u_0), & i = 1,2,\dots n \\ \sum_{i=1}^{n} \lambda_i = 1 \end{cases} \quad (4)$$

where $C(u_i, u_j)$ is the fitted covariance function value between each pair of gauged
basins ($i$=1,…n), and $C_0(u_i, u_0)$ is the fitted covariance value between the location of
interest $u_0$ and each of the samples $u_i$, $\mu$ is the Lagrange multiplier. After calculating
the weights, $\lambda_i$, and substituting them into Eq. (3), the runoff prediction in the region of
interest is solved by the linear combination of the weights and the observed runoff.

To calculate the weights, we write Eq. (4) into a matrix form: $C\Lambda = C_0$, and the

weights matrix as
$$\Lambda = C^{-1} C_0 \quad (5)$$

where
$$C = \begin{bmatrix} Cov(u_1, u_1) & Cov(u_1, u_2) & \cdots & Cov(u_1, u_n) & 1 \\ Cov(u_2, u_1) & Cov(u_2, u_2) & \vdots & Cov(u_2, u_n) & 1 \\ \vdots & \vdots & \vdots & \vdots & \vdots \\ Cov(u_n, u_1) & Cov(u_n, u_2) & \cdots & Cov(u_n, u_n) & 1 \\ 1 & 1 & 1 & 1 & 0 \end{bmatrix} \quad (6)$$

$$C_0 = \begin{bmatrix} Cov(u_1, u_0) \\ Cov(u_2, u_0) \\ \vdots \\ Cov(u_n, u_0) \\ 1 \end{bmatrix} \quad (7)$$

$$\Lambda = \begin{bmatrix} \lambda_1 \\ \lambda_2 \\ \vdots \\ \lambda_n \\ \mu \end{bmatrix} \quad (8)$$

Eq. (4) is the main equation set of the stochastic interpolation approach. In the runoff
interpolation procedure, the fundamental unit is the block instead of point, thus matrix
$C$ represents covariance function value of the pair blocks, and matrix $C_0$ is the





covariance of block and the location of interest. The covariance values are function of
block, not spatial location. Eqs. (3) and (4) only present one location to be predicted. If
the interpolation procedure is multiple $M$ non-overlapping sub-basins, Eq. (5) will be the
same, but the optimal weights must be solved using the following set of equations
(Sauquet and Gottschalk, 2000):
$$\Lambda = \begin{bmatrix} L_1 \\ L_2 \\ \vdots \\ L_M \\ \mu^* \end{bmatrix} \text{ and } L_i = \begin{bmatrix} \lambda_1^i \\ \lambda_2^i \\ \vdots \\ \lambda_n^i \\ \mu^i \end{bmatrix} \tag{9}$$

where $L_i$ is the weights of all the sample observations with respect to the $i - th$ sub-
basin. The matrixes $C$ and $C_0$ in Eqs. (6) and (7) are
$$C = \begin{bmatrix} K & 0 & \cdots & 0 & V_1 \\ 0 & K & 0 & \vdots & V_2 \\ \vdots & 0 & \ddots & 0 & \vdots \\ 0 & \cdots & 0 & K & V_M \\ V_1^T & V_2^T & \cdots & V_M^T & 0 \end{bmatrix} \text{ and } C_0 = \begin{bmatrix} G_1 \\ G_2 \\ \vdots \\ G_M \\ n_T q_T \end{bmatrix} \tag{10}$$

In Eq. (10)
$$K = \begin{bmatrix} Cov(A_1, A_1) & Cov(A_1, A_2) & \cdots & Cov(A_1, A_n) & 1 \\ Cov(A_2, A_1) & Cov(A_2, A_2) & \cdots & Cov(A_2, A_n) & 1 \\ \vdots & \vdots & \cdots & \vdots & \vdots \\ Cov(A_n, A_1) & Cov(A_n, A_2) & \cdots & Cov(A_n, A_n) & 1 \\ 1 & 1 & 1 & 1 & 0 \end{bmatrix} \tag{11}$$

and
$$V_i = \begin{bmatrix} n_i r(A_1) \\ n_i r(A_2) \\ \vdots \\ n_i r(A_n) \\ 0 \end{bmatrix} \tag{12}$$

$$G_i = \begin{bmatrix} Cov(A_1, \Delta A_i) \\ Cov(A_2, \Delta A_i) \\ \vdots \\ Cov(A_n, \Delta A_i) \\ \mu^i \end{bmatrix} \tag{13}$$

where $\Delta A_i$ is the non-overlapping area for sub-basin $i$ $(i = 1, \dots M)$.





Unlike the random point interpolation, the above set of matrix equations should be
constrained by water balance, i.e., the sum of the interpolated discharge for each sub-
basin should equal to the observed discharge in its river outlet. This constraint equation
can be expressed specifically as
$$R_T = \sum_{i=1}^{M} \Delta A_i \, r(\Delta A_i) \qquad (14)$$
where $R_T$ is total streamflow observed at outlet of the basin.
On grid estimation of runoff (Sauquet and Gottschalk, 2000), each of the non-
overlapping areas $\Delta A_i$ is further subdivided into $n_i$ grids surrounded by an area of $a$.
The runoff prediction of each $\Delta A_i$ is the linear combination of weights and runoff
observations presented as Eq. (15). Rearrange Eq. (14) yields
$$R_T = \sum_{i=1}^{M} \left( \sum_{j=1}^{n} n_i \, \lambda_j^i r(A_j) \right) = n_T r_T \qquad (15)$$
where $n_T$ is the number of fundamental grids; and $r_T$ is the runoff depth in outlet of
the basin.
To develop the theoretical covariance function $C$ and then the matrixes $C_0$ and $G_0$,
the fundamental step is to define the distance between a pair of sub-basins from the
identified runoff hierarchical structure in the river system. The appropriate geostatistical
distance between sub-basins A and B defined by Gottschalk (1993b) is expressed as the
expectation of distances of all the possible sub-basin pairs:
$$d(A, B) = \frac{1}{A_1 A_2} \int \int_{A_1 A_2} ||u_1 - u_2|| du_1 du_2 \qquad (16)$$
where $A_1$ and $A_2$ are the areas of sub-basin A and B.
Based on the sub-basin distance, an empirical covariogram versus geostatistical
distance can be drawn in a scatter diagram. The theoretical covariogram $Cov(A, B)$ is



derived in the same way as geostatistical distance
$$Cov(A, B) = \frac{1}{A_1 A_2} \int \int_{A_1 A_2} Cov_p(||u_1 - u_2||) du_1 du_2 \qquad (17)$$

In the above, $Cov_p$ is the point covariance function and can be calibrated by trial-and-
error fitting method. Only independent drainage basins are used to calculate the
covariance function to avoid spatial correlation of the nested drainage basins.

**2.3 Hydro-stochastic interpolation scheme with Budyko approach**
The above stochastic interpolation procedure assumes a stationary stochastic
variation of runoff among sub-basins or spatial homogeneity in runoff features (Sauquet,
2006) despite consideration of the river network structure. However, nonstationary
variation of runoff from spatial heterogeneity in the river system often exists due to
distinguishing spatial variability in climate-landscape factors, such as regional
distribution of rainfall, evapotranspiration, topography and soils, particularly in large
basins. Thus, the spatial runoff can be decomposed into nonstationary deterministic and
stochastic components:
$$R(x) = R_d(x) + R_s(x). \qquad (18)$$

In (18), $R(x)$ is runoff at location $x$, $R_d(x)$ is the deterministic component of the
spatial trend and/or the external drift (Wackernagel, 1995) that results in nonstationary
variability, $R_s(x)$ is the stochastic component regarded as a stationary variable.
In this study, Fu's equation (Eq. (1)) is used as an external drift function, $R_d(x)$ in
(18), accounting for the deterministic variation of mean runoff in space. The residual
$R_s(x)$ [the sub-basin runoff $R(x)$ minus the external drift $R_d(x)$] is used for executing





the hydro-stochastic interpolation scheme known as ''residual kriging'' (Sauquet, 2006).
The sum of $R_s(x)$ and $R_d(x)$, i.e., $R(x)$ predict of runoff at ungauged sub-basins.
**2.4 Cross validation**

The validation procedure for (18) is conducted using leave-one-out cross-validation

method (Kearns, 1999) in order to examine and compare quantitatively the performances
of three prediction models (Budyko approach, hydro-stochastic interpolation, and
coupling). The performance of each model is evaluated by the same metrics (Laaha and
Bloschl, 2006):
$$\text{MAE} = \frac{1}{n}\sum_{j=1}^{n}[R(x_i) - R^*(x_i)]$$    (19)
$$\text{MSE} = \frac{1}{n}\sum_{j=1}^{n}[R(x_i) - R^*(x_i)]^2$$    (20)
$$\text{RMSE} = \sqrt{\frac{1}{n}\sum_{j=1}^{n}[R(x_i) - R^*(x_i)]^2}$$    (21)
where $R^*(x)$ is the prediction of spatial variable $R(x)$. *MAE* is mean absolute error,
*MSE* is mean square error, *RMSE* is root-mean-square error between prediction values
and observation data.

The coefficient of determination for cross-validation is

$$\text{R}_{cv}^2 = 1 - \frac{V_{cv}}{V_{NK}}$$    (22)
where $V_{cv}$ is mean square error (*MSE*); and $V_{NK}$ is the spatial variance of the runoff
over all the tested sub-basins.

The prediction result can be illustrated by regression analysis of the observation vs.

prediction in addition to the evaluation metrics and $R_{cv}^2$.
**3. Study catchment and data**

Huaihe River Basin (HRB), the sixth largest basin in China, was selected for the





validation of the spatial interpolation of runoff. HRB is of particular interest because of
its situation in China's transition terrain from north to south, and the transition climate
from the warm temperate monsoonal climate in the east to sub-humid climate on the
west (Hu, 2008). The basin has the most human population density, and is one of the
major agricultural areas in China. Millions of tons of water are consumed each year to
sustain the population and agriculture. Water resources per capita and per unit area is less
than one-fifth of the national average. Moreover, more than 50% of the water resources
is overexploited, much higher than the recommended rate for international inland rivers
(30%) (Yan et al, 2011). Higher precipitation concentration, represented by large
percentages of the annual precipitation in a few very rainy months, makes the region
vulnerable for severe floods as well as droughts. The frequent droughts and floods
increase difficulty in water resources utilization and flood prevention (Zhang et al.,

2015).

The selected study area is located upstream of Bengbu Sluice in HRB with an area
of 121,000 km$^2$ (Fig. 1). The river network system is derived from data packages of
National Fundamental Geographic Information System issued by National Geomatics
Center of China. The area in the upstream is divided into 40 sub-basins, in terms of
available hydrological stations with records within the period 1961-2000 (Fig. 2). The
sub-basin area varies from the smallest of 17.9 km$^2$ (at PH station) to the largest of 30630
km$^2$ (WJB station). Among the 40 sub-basins, there are 27 independent sub-basins and
13 nested sub-basins of the observation network.
Annual precipitation data from 1961-2000 are obtained from monthly mean



climatological dataset at 0.5° spatial resolution constructed by China Meteorological
Administration (available at: http://data.cma.cn/data/detail/dataCode/SURF_CLI_CHN
_PRE_MON_GRID_0.5/keywords/0.5.html). Pan evaporation data at 21 meteorological
stations in HRB are used to interpolate spatial potential evapotranspiration via ArcGIS,
and then the annual potential evapotranspiration of each sub-basin in HRB is obtained.
The statistical features of mean annual precipitation, potential evapotranspiration and
runoff during the period from 1961-2000 are listed in Table 1. During 1961-2000, the
mean annual precipitation $P$ varied from 638~1629 mm; mean annual temperature was
11~16℃, and the mean annual potential evaporation $E_0$ varied between 900~1200 mm.
The sub-basins in the north are relatively dry with the dryness index ($E_0/P$) higher than
1.3 for the sub-basins of ZM, ZQ, XY and ZK. Sub-basins in the south are wet with the
dryness index ($E_0/P$) lower than 0.8 for the sub-basins of MS, HBT and HC. The average
mean annual runoff depth $R$ is about 400 mm, but fluctuating from a minimum of 90 mm
in the northern region near the Yellow River to a maximum of 1000mm in the south
mountainous areas. The temporal and spatial variation of runoff of HRB is relatively
small in the south but large in the north.

**4 Results**
**4.1 Prediction of runoff by Fu's equation**

Actual evapotranspiration $E$ (Table 1) is estimated according to long-term mean of

annual water balance ($E=P-R$). On the basis of Eq. (1) and long-term mean of annual
water balance components during 1961–2000 at the 40 sub-basins (Table 1), we plot the
$E/P$ vs. $E_0/P$ in Fig. 3. In Fig. 3, we also include the water limit line of the arid edge at



which $E = P$ and the energy limit line of the wet edge at which $E = E_0$. The curve shape
in Fig. 3 is determined by the parameter $\omega$. Its value in each sub-basin is calculated
directly using Eq. (1) and is listed in Table 1. The range of $\omega$ is from the smallest 1.43 at
HWH to the largest 3.16 at JJJ, the average of $\omega$ is 2.32 over the 40 sub-basins.
The sub-basin averaged $\omega$ can be fitted by minimizing the mean absolute error (*MAE*)
(Legates and McCabe, 1999) between the predicted and the estimated annual
evapotranspiration $E$ from the long-term water balance (Fig. 3). The fitted value of $\omega$ for
the 40 sub-basins determined from this process is 2.213, very close to the average
directly from the 40 individual sub-basins.
Using $\omega=2.213$ in our study basin, Fu's Eq. (2) is written
$$R = \left(1 + \left(\tfrac{E_0}{P}\right)^{2.213}\right)^{\frac{1}{2.213}} - E_0. \qquad (23)$$

Eq. (23) and Fig. 3 clearly show the deterministic trend of runoff in space. The smaller
the index $\frac{E_0}{P}$ is, the larger the runoff is over the sub-basins in HRB. In Fig. 3, the larger
$R$ in the sub-basins in the north indicates drier conditions in those sub-basins.
Using Eq. (23) and the mean annual precipitation $P$ and potential evapotranspiration
$E_0$ at the 40 sub-basins given in Table 1, the predicted runoff depth by Fu's equation and
deviation, or prediction error, between prediction and observation are calculated. The
results are also summarized in Table 1 and 2. The *MAE* of Budyko runoff prediction is
94 mm, and the *RMSE* is 112 mm. The largest absolute error is at HWH (328.03 mm)
and the smallest at XX (23.77 mm) (Table 1 and 2). The largest relative error is 91 mm
at XZ station, about 81.6% of the observed runoff at the site, and the smallest is 36.94
mm at XHD, 4.99% of the observed runoff.






### 4.2 Hydro-stochastic interpolation of runoff

For comparison, direct use of the observed runoff in the hydro-stochastic
interpolation is executed based on the procedure detailed in Section 2.2. The covariance
was firstly calculated by

$$C(d) = E[\, r(x_i) \cdot r(x_i + d)] - \bar{r}^{\,2} \qquad (24)$$

where $\bar{r}$ is the average of the observed runoff among the sub-basins, $d$ is the
geostatistical distance between pairs of the sub-basins.
In order to obtain the distance $d$ between the sub-basin pairs, HRB is divided into
grids of 40 row $\times$ 50 column resolution. According to Eq. (16), the geostatistical
distances of all the possible sub-basin pairs (820 in this study) were calculated to obtain
the average distance of each pair of grid points in sub-basins A and B. According to the
estimated distance for pairs of sub-basins and the observed runoff at 40 sub-basins (Table
1), the empirical covariance $C(d)$ is estimated for each pair of sub-basins. From plots
of the mean estimated $C(d)$ of the independent sub-basin pairs versus the
corresponding distances $d$ with an interval of 50 km, we get an empirical covariogram
shown in Fig. 4. The best fit to this empirical covariogram is

$$C(d) = 600000 \times \exp(-d/28.62). \qquad (25)$$

The fitted exponential function in (25) is used to calculate the theoretical covariance
matrix $Cov(A, B)$ in Eq. (17). Then the matrices of $C$, $C_0$, $K$, $V$ and $G$ are
subsequently generated by MATLAB programs, and the weight coefficient matrix is
calculated consequently.



The interpolation results over 40 sub-basins are conducted, and the prediction error
is shown in Table 1. The *MAE* and *RMSE* are 134 mm and 176mm, respectively. The
largest absolute error is at HWH (448 mm) and the smallest at XHD (3 mm) (Table 2).
The interpolation errors are larger than those from the Budyko curve, a result indicating
that the observed runoff is controlled by the deterministic trend in space, which markedly
affects spatial interpolation accuracy.

**4.3 Hydro-stochastic interpolation with Fu's equation**
Because of the significant deterministic trend of runoff in space, the trend removal
can help justify assumptions of spatially-autocorrelated random error for the hydro-
stochastic interpolation. Following Section 2.3, we use Fu's equation (Eq. (2)) to
estimate the deterministic trend or the external drift function $R_d^*(x)$, and departures of
the trend, or the residual/errors, between the prediction and observation. The residual
$R_s^*(x)$ is used for hydro-stochastic interpolation. The results are given in Table 1.
The empirical covariogram of $R_s^*(x)$ for each pair of sub-basins versus sub-basin
distances is plotted in Fig. 5. The following exponential function is obtained from the
best fitting the empirical covariogram
$$C(d) = 3000 \times \exp(-d/48.34). \tag{26}$$

From (26), matrices $C$, $C_0$, $K$, $V$ and $G$ in Eqs. (9) ~ (13) are calculated using
MATLAB, and the weight coefficient matrix of runoff deviation is then calculated to
predict runoff deviation. Since this interpolation scheme represents the spatial runoff
deviation, the sum of the interpolated runoff deviation and the simulated runoff by Fu's



equation is regarded as the total interpolated runoff in sub-basins.
Prediction outcome of runoff is listed in Table 1, with the *MAE* of 47 mm and *RMSE*
of 69mm over the 40 sub-basins. The largest absolute error is at HWH (236 mm) and the
smallest at JJJ (1.5 mm) (Table 2).
**4.4 Comparison of spatial runoff perdition by the three approaches**

As listed in Table 2, our method coupling the deterministic and stochastic processes

described in this study significantly reduces the prediction errors in space. *MAE* and
*RMSE* from the coupled method are much smaller than those from the Budyko and the
hydro-stochastic interpolation methods. The maximum error at HWH is significantly
reduced as well; 236 mm from the coupled method is much smaller than 328 mm from
the Budyko method and 448 mm from the hydro-stochastic interpolation. In terms of the
cross-validation outcome in Table 2, the cross-validation outcome $R_{cv}^2$ from our
coupled method is as large as 0.93, much larger than 0.81 and 0.54 from the Budyko
method and the hydro-stochastic interpolation, respectively.

The correlation analysis between predicted and observed runoff depth is shown in

Fig. 6. The prediction from our coupled method is highly correlated with the prediction
($R^2$=0.95) and small deviation from the 1:1 line. In contrast, correlation between the
prediction and observation from the Budyko method and the hydro-stochastic
interpolation is low ($R^2$=0.58 and 0.82, respectively). Particularly, they markedly
overestimate low runoff and underestimate high runoff (strong departures to 1:1 line in
Fig. 6). The systematic deviation of the runoff prediction by the hydro-stochastic
interpolation has also been reported in the previous work by Sauquet et al. (2000), Laaha



and Bloschl (2006) and Yan et al. (2011).
Mapping spatial distribution of runoff in HRB by the three approaches of Budyko
equation, hydro-stochastic interpolation and our coupled method is shown in Fig. 7.
There are significant differences in mapped runoff distribution in HRB by the three-
spatial interpolation methods. Compared with our coupled method, the Budyko method
and hydro-stochastic interpolation markedly underestimate sub-basin runoff in the north
where climate is relatively dry and runoff is small. Among the predicted runoff in the
largest non-overlapping area above BB, the one made by our coupled method is 125mm,
and the ones made by the Budyko method and the hydro-stochastic interpolation are 264
and 179 mm, respectively.
**5. Discussion and conclusions**
Investigating the underlying patterns of hydrological variables is important in our
effort to obtain good knowledge of spatial variation of the hydrological variables in a
region of interest. Because of existence of some degree of natural organization or
connection of water basins (Dooge, 1986; Sivapalan, 2005), e.g., rivers that connect sub-
basins, and hydro-climate similarity, we can describe the hydrological variables of
interest in deterministic forms of functions, curves or distributions and construct
conceptual and mathematical models to predict hydro-climate variability (Wagener et al,
2007). However, the deterministic method in describing complex patterns suffers
inevitable loss of information (Wagener et al, 2007) because of existence of uncertainty
in many hydrological processes and data. Thus, hydrological variables also contain
information of stochastic nature, and should be treated as outcomes of both deterministic
and stochastic processes. Use of combined or coupled deterministic and statistical





hydrological models to predict hydrological processes has been recommended and
proven to be effective in improving accuracy of various aspects of hydrology, such as
hydrological forecast (Cheng et al., 2014; Ly et al., 2013) and groundwater table
interpolation (Holman et al., 2009).

In this study, we use the Budyko's deterministic method to describe mean annual

runoff as an integrated spatial continuous process determined by both the hydro-climate
elements and the hierarchical river system networks. A deviation from the Budyko
estimation is our use of the hydro-stochastic interpolation that assumes spatially-
autocorrelated random error. The predicted runoff is the coupling of predictions by the
Budyko method and the hydro-stochastic interpolation. The deterministic aspects of
runoff described by Budyko method reflect regional trends at positions (sub-basins) and
their deviations caused by stochastic processes are determined by the weights as a
function of physical distance. Weights are higher for near points/basins and are smaller
for distant points/basins.

We tested our coupled method in the Huaihe River basin in China. Our results show

that the coupled method outperformed both the Budyko method and the stochastic
interpolation by significantly increasing the spatial interpolation and prediction accuracy.
The interpolation errors in terms of *MAE* and *RMSE* from our coupled method are 47
and 69mm over the 40 sub-basins, respectively, much smaller than 94 and 112 mm from
the Budyko, and 134 and 176mm from the hydro-stochastic interpolation. The maximum
error at HWH is significantly reduced as well. It is 236 mm from our coupled method
much smaller than 328 mm from the Budyko method and 448 mm from the hydro-



stochastic interpolation. The cross-validation outcome of the deterministic coefficient
$R^2_{cv}$ from our coupled method is 0.93, much larger than 0.81 and 0.54 from the Budyko
method and the hydro-stochastic interpolation, respectively. The prediction from the
coupled method captures most accurately among the three methods the regional high and
low runoff in the HRB.
Because our coupled method incorporates climate conditions, e.g., precipitation and
evapotranspiration, it provides a useful tool to estimate climate change impacts on long-
term water availability in large-scale river basins and to assess potential consequences
of climate change in environment and water and food security.

**Acknowledgement**
The research was supported by the National Natural Science Foundation of China
(No. 51190091 and 41571130071).

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

interpolation; (c) coupled method






Table 1 Summary of hydro-meteorological data and predicted runoff of sub-basins in HRB

| No | Stations | Basin area\km² | P (mm) | R (mm) | E₀ (mm) | E₀/P | E (mm) | ω | Fu's Equation | | Hydro-stochastic interpolation | | Coupling method | |
|---|---|---|---|---|---|---|---|---|---|---|---|---|---|---|
| | | | | | | | | | Predicted R (mm) | Error (mm) | Predicted R (mm) | Error (mm) | Predicted R (mm) | Error (mm) |
| 1 | CTG | 3090 | 1012 | 366 | 932 | 0.92 | 646 | 2.41 | 399 | 32.85 | 371 | 4.90 | 348 | 17.84 |
| 2 | XHD | 1431 | 1517 | 740 | 974 | 0.64 | 776 | 2.41 | 777 | 36.94 | 737 | 2.70 | 692 | 47.82 |
| 3 | SQ | 3094 | 822 | 168 | 1024 | 1.25 | 653 | 2.83 | 248 | 79.29 | 285 | 116.77 | 178 | 10.10 |
| 4 | MS | 1970 | 1517 | 672 | 957 | 0.63 | 845 | 3.06 | 786 | 114.28 | 584 | 88.45 | 662 | 10.13 |
| 5 | BGS | 2730 | 877 | 225 | 1029 | 1.17 | 651 | 2.57 | 279 | 53.93 | 247 | 22.39 | 181 | 44.01 |
| 6 | XC | 4110 | 945 | 225 | 997 | 1.06 | 720 | 3.02 | 332 | 106.82 | 272 | 46.77 | 212 | 13.11 |
| 7 | BT | 11280 | 910 | 223 | 993 | 1.09 | 687 | 2.85 | 310 | 86.94 | 275 | 52.25 | 219 | 3.74 |
| 8 | ZK | 25800 | 678 | 123 | 1061 | 1.56 | 555 | 2.54 | 163 | 39.96 | 228 | 104.65 | 61 | 61.70 |
| 9 | JJJ | 5930 | 1347 | 513 | 969 | 0.72 | 834 | 3.16 | 640 | 127.27 | 520 | 7.49 | 512 | 1.49 |
| 10 | HB | 16005 | 1092 | 335 | 937 | 0.86 | 757 | 3.15 | 455 | 120.48 | 334 | 1.02 | 360 | 25.01 |
| 11 | ZQ | 3410 | 739 | 118 | 1083 | 1.47 | 621 | 2.83 | 190 | 71.71 | 219 | 101.07 | 141 | 23.40 |
| 12 | HPT | 4370 | 1629 | 764 | 984 | 0.60 | 865 | 2.92 | 868 | 103.53 | 755 | 9.22 | 712 | 51.64 |
| 13 | XX | 10190 | 987 | 367 | 1053 | 1.07 | 620 | 2.10 | 343 | 23.77 | 381 | 13.73 | 424 | 56.96 |
| 14 | BB | 121330 | 850 | 215 | 1024 | 1.20 | 635 | 2.54 | 264 | 49.48 | 394 | 179.16 | 125 | 90.46 |
| 15 | WJB | 30630 | 1003 | 294 | 957 | 0.95 | 709 | 2.85 | 384 | 90.29 | 304 | 9.65 | 287 | 6.90 |
| 16 | LZ | 390 | 963 | 345 | 1078 | 1.12 | 618 | 2.09 | 320 | 24.96 | 320 | 25.08 | 399 | 53.75 |
| 17 | NLD | 1500 | 1019 | 439 | 1101 | 1.08 | 581 | 1.86 | 351 | 88.30 | 309 | 129.64 | 401 | 37.56 |
| 18 | ZMD | 109 | 690 | 212 | 1093 | 1.58 | 478 | 1.94 | 163 | 48.65 | 281 | 68.78 | 235 | 22.53 |
| 19 | BLY | 737 | 1504 | 868 | 1126 | 0.75 | 635 | 1.69 | 695 | 173.27 | 639 | 229.05 | 794 | 74.23 |
| 20 | HWH | 292 | 1560 | 1068 | 1127 | 0.72 | 492 | 1.43 | 740 | 328.03 | 619 | 448.83 | 832 | 236.16 |
| 21 | ZC | 493 | 1512 | 838 | 1112 | 0.74 | 674 | 1.79 | 708 | 130.23 | 695 | 142.77 | 777 | 61.19 |
| 22 | BQY | 284 | 1268 | 693 | 1094 | 0.86 | 575 | 1.68 | 527 | 166.21 | 349 | 344.06 | 604 | 89.35 |
| 23 | QL | 178 | 1559 | 970 | 1090 | 0.70 | 589 | 1.60 | 756 | 214.17 | 646 | 324.06 | 840 | 130.17 |
| 24 | HNZ | 805 | 1480 | 640 | 1114 | 0.75 | 840 | 2.41 | 681 | 41.37 | 577 | 63.05 | 585 | 55.20 |
| 25 | TJH | 152 | 1305 | 699 | 1090 | 0.84 | 605 | 1.74 | 556 | 143.66 | 262 | 437.02 | 589 | 110.18 |
| 26 | LX | 77.8 | 1025 | 484 | 1079 | 1.05 | 540 | 1.75 | 361 | 123.77 | 241 | 242.88 | 436 | 48.01 |
| 27 | ZLS | 1880 | 755 | 253 | 1104 | 1.46 | 502 | 1.91 | 194 | 58.45 | 169 | 84.28 | 233 | 19.94 |
| 28 | ZT | 501 | 1021 | 437 | 1101 | 1.08 | 583 | 1.87 | 351 | 85.87 | 242 | 195.10 | 411 | 26.08 |
| 29 | XGS | 375 | 830 | 302 | 1088 | 1.31 | 528 | 1.91 | 238 | 63.74 | 243 | 58.60 | 297 | 5.46 |





| 30 | JZ | 46 | 1103 | 583 | 1107 | 1.00 | 520 | 1.63 | 404 | 178.81 | 200 | 382.51 | 455 | 127.50 |
| 31 | GC | 620 | 638 | 111 | 1055 | 1.65 | 528 | 2.51 | 145 | 34.18 | 139 | 28.42 | 103 | 8.08 |
| 32 | ZM | 2106 | 645 | 97 | 1039 | 1.61 | 548 | 2.72 | 150 | 53.48 | 141 | 43.80 | 105 | 7.58 |
| 33 | YZ | 814 | 979 | 235 | 1083 | 1.11 | 743 | 2.85 | 329 | 94.07 | 277 | 42.13 | 246 | 11.24 |
| 34 | XZ | 1120 | 746 | 111 | 1040 | 1.39 | 636 | 3.06 | 202 | 90.66 | 167 | 56.30 | 152 | 40.95 |
| 35 | GZ | 1030 | 855 | 342 | 1098 | 1.28 | 513 | 1.81 | 250 | 92.10 | 255 | 86.54 | 307 | 35.14 |
| 36 | DPL | 1770 | 1067 | 331 | 1066 | 1.00 | 736 | 2.57 | 393 | 61.62 | 339 | 8.02 | 342 | 11.39 |
| 37 | XX2 | 256 | 1301 | 606 | 1092 | 0.84 | 695 | 2.00 | 552 | 53.68 | 705 | 99.36 | 552 | 53.82 |
| 38 | PH | 17.9 | 1248 | 708 | 1094 | 0.88 | 540 | 1.61 | 512 | 196.04 | 604 | 104.35 | 512 | 195.9 0 |
| 39 | HC | 2050 | 1255 | 454 | 1095 | 0.87 | 802 | 2.54 | 517 | 63.36 | 363 | 91.02 | 409 | 44.52 |
| 40 | HK | 2141 | 871 | 227 | 1077 | 1.24 | 644 | 2.44 | 264 | 37.28 | 309 | 82.40 | 186 | 41.22 |








Table 2 Interpolation and cross-validation errors between the predicted and observed

runoff at 40 sub-basins for the three methods

| Evaluation indicators | Budyko | Hydro-stochastic interpolation | Coupling method |
|---|---|---|---|
| MAE (mm) | 94 | 134 | 47 |
| RMSE (mm) | 112 | 176 | 69 |
| Max error (mm) | 328 | 448 | 236 |
| Min error (mm) | 24 | 3 | 1.5 |
| $R^2_{cv}$ | 0.81 | 0.54 | 0.93 |








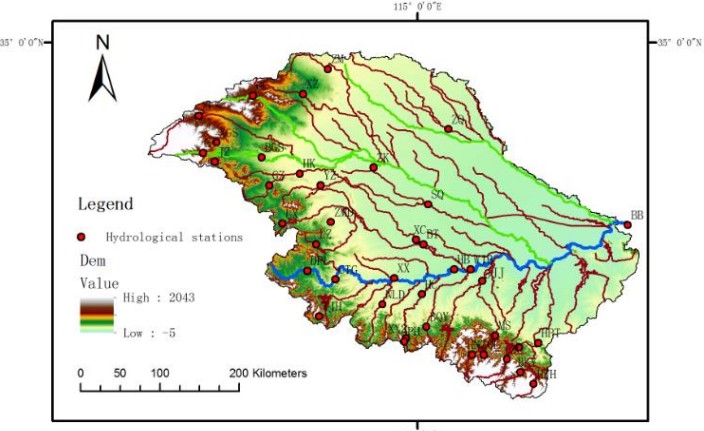


Figure 1 Topography and river network of HRB above Bengbu


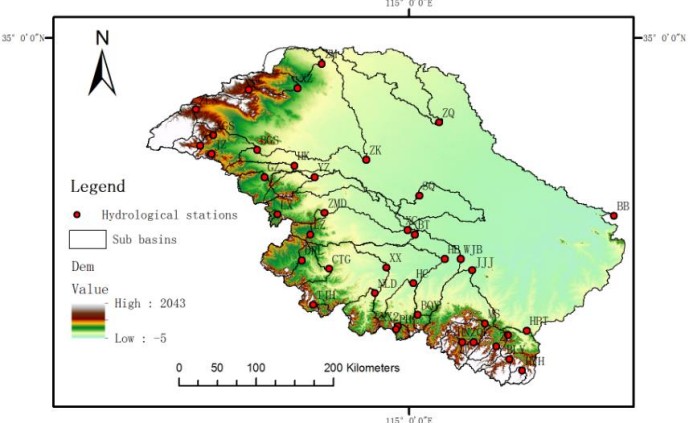


Figure 2 Sub-basins and hydrological stations of HRB above Bengbu







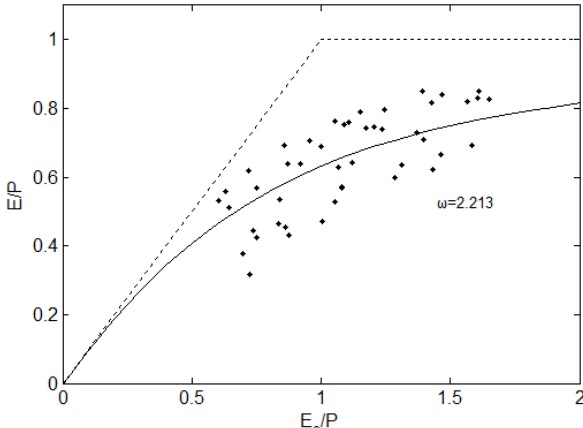


Figure 3 Plot of *E/P ~E₀/P* for 40 sub-basins and Budyko curve of HRB






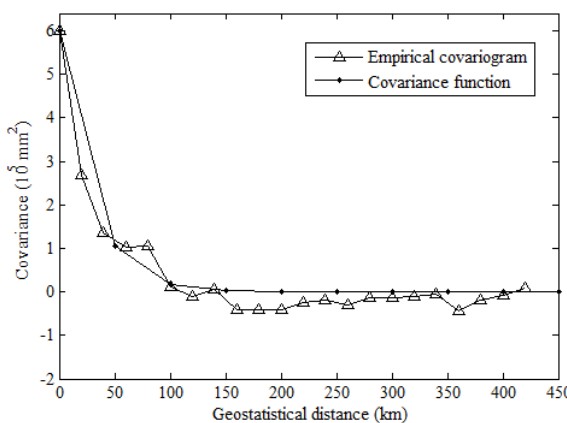


Figure 4 Empirical covariogram from sub-basin runoff data and fitted covariogram of

HRB


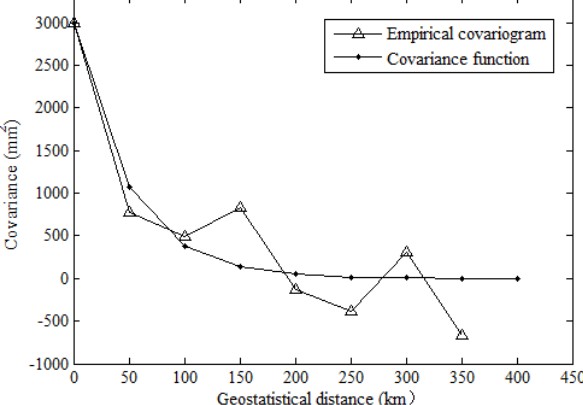


Figure 5 Empirical covariogram from sub-basin runoff deviation and fitted

covariogram of HRB



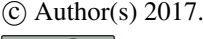




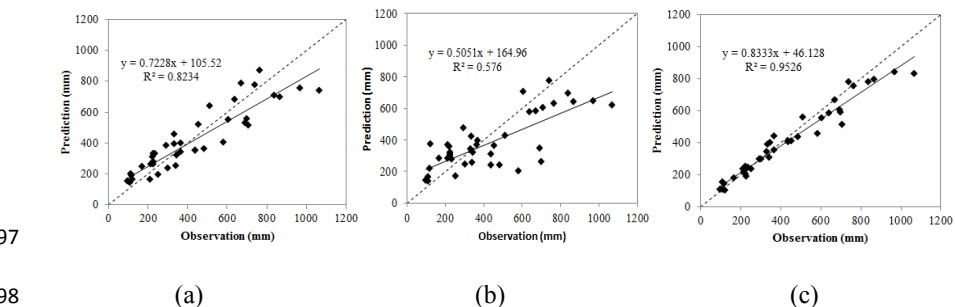


          (a)                      (b)                      (c)


Figure 6 Cross validation of runoff prediction vs. observation by (a) Fu's equation, (b)

hydro-stochastic interpolation, and (c) coupled method. The dotted line is 1:1


(a)                                          (b)

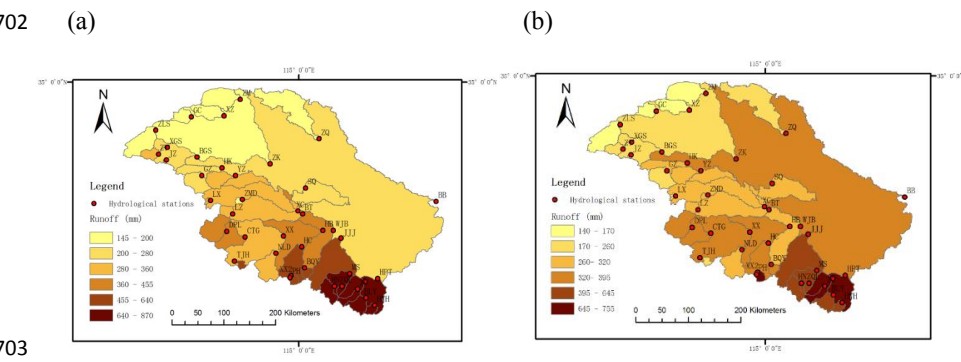


(c)

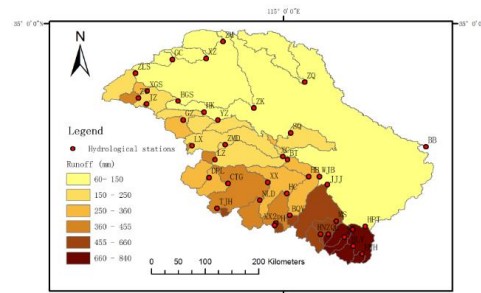


Figure 7 Spatial distribution of mean annul runoff: (a) Budyko; (b) hydro-stochastic

interpolation; (c) coupled method