# Peer review of "Hydro-stochastic interpolation coupling with Budyko approach for prediction of"

_Hydrology and Earth System Sciences, 2017_

## Referee Comment (RC1) · M. Mälicke (Referee) · 27 Oct 2017

**Summary**

I would like to thank the authors for this interesting contribution, I appreciated reading it. The authors presented a new interpolation approach for mean annual runoff depths. It was presented as a proof-of-concept comparing the interpolation results to long-term observations in a catchment in China.
The proposed method is, to my understanding, an extension of the commonly used estimation approaches based on the Budyko curve. The authors describe these approaches as semi-empirical, or deterministic, and try to account for an assumed

stochastic error on runoff depth observations by using a geostatistical interpolation method to regionalize errors observed in gauged basins to ungauged basins. By describing deterministic shares of runoff by the Budyko curve and utilizing observed covariance structures in the deviations, the authors could show, that the combination of both can yield significant better results, than each method on its own.

**Evaluation**

In my opinion, the proposed method is of relevance and adds value to recent issues in runoff depth interpolation. The results seem promising and the methods were presented largely in a clear and transferable way. Except from some technical remarks, the figures were relevant and descriptive.
I would recommend to propose some major revisions for the methods, as these need clarification or extension in some parts. The results were presented in figures, tables and in word, where I would propose to add more numbers in order to obtain an even more comprehensive insight. A major revision is also proposed for the discussion, as this section does only summarize the results in most parts. My remarks for revising this work can be found below.

I kindly ask to consider my remarks and finally I would like to see this work published in HESS.

**Major points**

- l.63 – 71: This paragraph consists of only two sentences, which are way too long and thus, were confusing for me. In the first sentence the authors make two different points. First, streamflow is a combined landscape information and second, that climate-landscape variability leads to non-stationary runoff observations. I kindly ask the authors to separate these points and reword the following statements in order to foster the structure. Especially the term "deterministic term"

(l. 65) needs more and clearer introduction. This is in the following work also referred to as "deterministic trend" and is of fundamental importance for the proposed method. Introducing this term in more detail will significantly increase my text comprehension for the entire work.

The second sentence in this paragraph (l. 68 – 71) does in my opinion not connect to the first one and it was not obvious what this sentence shall emphasize. What trend does the "the spatially nonstationary trend of runoff" (l. 68) refer to? And how is a runoff trend interpretable as "hydrological regionalization in terms of hydro-climate and landscape data" (l. 68 – 69)? What I read out of this sentence is that non-stationary runoff is caused by heterogeneity in hydro-climate and the landscape and can be described by empirical relationships as done in the presented studies (l. 71). But this is not exactly what is written down in this paragraph.

In my opinion, the authors shall rewrite the whole paragraph in shorter, non-nested sentences.

- I would strongly recommend to completely rework the whole section 5 from line 408 to 451, due to many factors. Above all, this whole section is neither a discussion nor a conclusion in my opinion.

  The first paragraph (l.409-424) basically lays the framework for coupling "deterministic and statistical models" (l.420), which is used as a justification for the proposed method. The paragraph itself seems to be helpful and relevant but should thus be moved to the introduction, somewhere located (and linked) to the paragraph l.105-122.

  This paragraph is followed by two paragraphs that summarize major parts of the publication. l.425-434 summarizes the proposed method; while l.435 - 447 summarizes the reported result.

  The only conclusions drawn can be found in the last paragraph (l.448-451). In my opinion, these conclusion are way too general. Furthermore the authors presented a new interpolation method, while long-term climate change impacts are modeled into the future, which would require an extrapolation. Thus, the proposed method is not appropriate to predict climate change impacts.
As the authors presented some interesting results in this publication, it should be easy to draw some more immediate and definite conclusions.

- l.136 – 141: To me it is not clear why the authors have chosen Fu's equation. In the introduction to Budyko approaches (l. 129 – 136) the authors introduced a number of adjustments and improvements to the original approach suggested by other studies and highlighted their importance. Fu's equation does not incorporate any of these, but rather a "dimensionless model parameter" (l. 144), which does only control the "partitioning of precipitation into runoff" (l. 145). The authors are kindly asked to give more insights on this decision. Additionally, the calibration of this parameter is just mentioned in l. 146, but not further described.

- l.240 - 244: For my understanding, this is the key paragraph of the methodology as it describes the actual coupling of Budyko with hydro-stochastic interpolation. I would summarize this as: 1.: $R_d(x)$ in equation (18) is substituted with equation (2) by setting $R_d(x) = R.$ and calculated for all basins. 2.: The residuals between $R_d(x)$ and observed $R$ is calculated for all gauged basins. Further, these residuals are interpolated for all ungauged basins by "residual kriging" (l.243). and set as $R_s(x)$ 3.: Equation (18) applies as the final result of this study.
Following the cited "residual kriging" from Sauquet (2006) it was not clear to me, how exactly the "residual kriging" is performed on the ungauged basins. The residuals from this study would be described by a first oder polynomial ("Accounting for spatial heterogenity", last paragraph, in Sauquet (2006)), and be combined with $\xi_q$, the error in residuals. But, for me, it is not clear how this $\xi_q$ or the $g$ from Sauquet (2006) were calculated. From my point of view, the interpolation scheme described in Sauquet (2006) seems to be closely related to the general approach presented by the authors. Then, the delimitation between the

two studies was not clear to me from the introduction. In any case a clarification of how $R_s(x)$ is calculated, how section 2.2 sets in and is linked here would be highly appreciated.

- l.219 - 220: Which scatter diagram are you referring to, here? Furthermore I can hardly imagine how such a diagram would look like. For my understanding, an empirical covariogram relates the separating distances of lag classes the the inner-class covariance observed in the data. Please describe how a diagram like this shall be scattered over the distances between all sub-basin combinations. Furthermore, equation (17) presented in line 222 is used to derive a theoretical covariogram. From my understanding (and in fact I am not sure what $u_1, u_2, du_1, du_2$ are referring to here, see minor point below) this will yield a single value $Cov(A, B)$ for sub-basins A and B. Does the theoretical covariogram then relate $Cov(A, B)$ to $d(A, B)$ defined in (16) (l.217)? If so, a more descriptive and clear explanation in the respective paragraph would be highly appreciated. Additionally, do $Cov(A, B)$ (l.220) and $Cov(u_i, u_n)$ (l.178-180) describe the same thing?

- The authors should consider to report their result more consistently and comprehensive. Beside a cross-validation, the authors compare the three different interpolation approaches by comparing the errors each method yielded. This error reporting in line 377-379; 355-356 and 328-331 shall be harmonized and report the same numbers.
I would suggest reporting the overall minimum, maximum and mean error found in a single sub-basin, along with the minimum, maximum and mean relative error (as share of basin-specific runoff) found in any sub-basin. Both kind of errors can be reported as a absolute (in mm) and relative (in %) number. In my opinion this makes sense as, for example, the sub-basin yielding the biggest absolute error in equation 2 (which is HWH), does not show the biggest relative error (as eg. SQ shows a bigger relative error).

Beside reporting these important numbers, the authors should consider to report these numbers in table 2, as well.

**Minor points**

- l.322 - 323: This observation is not supported by fig. 3. From my point of view it is not possible to derive the location of a sub-basin from this figure.

- l.83 – 88: The authors make different points here within one long confusing sentence. They are kindly asked to break this sentence down to the core statements of: 1.: runoff is an integrated spatial continuous process, not a field like precipitation; 2.: runoff interpolation must take the stream network into account; 3.: the stream network constraints the water balance up- and downstream.
  Furthermore, please clarify the connection between a water balance constraint and assumed runoff properties that can be traced back to field properties.

- l.90 – 91: Please explain "lateral streamflow" (l. 90). What is that and how is it connected to the topic? None of the two presented studies, that shall explain the link between runoff overestimation and "neglecting lateral streamflow" contain the term "lateral streamflow". Please clarify what the two studies actually indicate.

- l.92 – 96. For my understanding, this part is not linked to the other parts of this paragraph or the introduction as far as I read it at that point. Why is this important? Additionally, "hydro-stochastic interpolation" (l. 92) was not clear to me at that point and the authors might consider some more explanation.
  Furthermore, the difference between "Euclidean distances" (l. 94) used in "conventional stochastic methods" and the "spatial distance" (l. 95) is too vague for me. Consider adding an explanation.

- Please clarify what "sample" refers to in line 171.
- l. 362-364: The authors are asked to clarify what "trend removal" refers to here, as no kind of trend removal was reported in the methods. From that, what kind of assumptions do you "justify" from applying a trend removal? Do you assume the residuals to be spatially autocorrelated or do you assume an existing spatial autocorrelated random error underlying the residuals themselves, as the "hydro-stochastic interpolation" is performed on the residuals?
  Consider extending the corresponding methods part.

- is the $du_1, du_2$ used in (17) (l.222) and (18) (l.236) the same thing, or does the $d$ from (17) refer to the $d(A, B)$ calculated in (16) (l. 217)? If not, what is (16) then used for? If yes, please clarify the difference of the two used $du_1, du_2$.

- Please describe what "spatial variance" (l. 259) exactly means here and how it is defined.

- The used precipitation data is described to be a "climatological dataset" (l.287). What kind of data product is this? An interpolated and aggregated map from a observation network? A radar product?

- l.290 How was this interpolation conducted? "ArcGIS" is capable of more than one interpolation method. Please name the method, not the tool.

- What is the "relative error [of] 91 mm"? Is this the absolute error at XZ station, where the relative error is the largest observed of 81.6%?

- l. 340 - 348: Why was HRB divided into a grid? The corresponding methodological description of these results (l. 212 - 217) did not mention this step. Furthermore, for me the link between equation 24 (l.337), equation 25 (l. 349) and figure 4 is not clear. Both equations describe a empirical covariance $C(d)$, while figure 4 shows a "covarinance function" along with an "empirical covariogram". Which one does refer to what here? The authors are kindly asked to make this clearer and the notation more distinct.

- l.351 How shall equation 25 be used to "calculate the theoretical covarinace matrix $Cov(A,B)$"? In line 220 $Cov(A,B)$ was described as a "theoretical covariogram", not a matrix. Is $Cov_p$ in equation 17 than the same as $C(d)$ in equation 25? Is the $d$ in equation 25 then derived from equation 16 for each sub-basin pair A,B? Are $u_1, u_2$ in equation 16 then the grid points mentioned in l.340 - 348 or the "samples" mentioned in line 171? Clarifying this specific step in the methods wherever appropriate would be highly appreciated.

- l.403-404 Did you mean that figure 7 (a) and (b) overestimate runoff, instead of underestimate, as stated? Because (a) ranges from 145mm - 280mm in the north and (b) ranges from 140mm - 280mm, in contrast (c) ranges from 60mm to 250mm in the north. Adding another sub-figure to figure 7 showing the measured runoff values can make figure 7 even more meaningful. Additionally, I would strongly recommend using the same value ranges for the color codes in figure 7, this will make the sub-figures more comparable and consistent.

**Technical points**

- In my opinion all the figures should be revised. The figure captions shall be extended and describe all figure elements. This is especially true for figures 3,4,5 and 6. Consider adding legends to figures 3 and 6.

- The authors are kindly asked to revise all their equations. Please make sure, that all used symbols are explained beneath the equation. This is especially true for $\mu^*$ and $\mu^i$ (l. 189); The sub- or superscripted $T$ used in e.g. in l 192; the undefined symbols $u_1, u_2, du_1, du_2$ (l.217); $Cov(u_i, u_n)$ in l.178-179, 194-197) . Wherever possible the symbol description shall also include the used unit. The unit was only given in a single case.

- l.129 – 136: This part is in fact a literature review on Budyko approaches and should thus be moved from the methodology part into the introduction.

- l.334 - 339: In my opinion, these are methods and should be moved to the correct section.

- l.314 - 316: Consider moving this to the methods (l.147-148), where the "calibration" is not further described.

- What exactly is meant by "drainage basin" in line 224? In the preceding text the authors referred to basins and sub-basins.

- Consider replacing "method with semi-empirical approach" (l. 112) with "method with semi-empirical Budyko approach", in order to be even more clear here.

- l.405 The authors should consider replacing "area above BB" with "area upstream of BB" or "area south of BB", to be more precise here.

- In line 388, I would not state that "0.93 [is] much larger than 0.81 and 0.54", as 0.93 - 0.81 is in fact smaller than 0.81 - 0.54. I would rather sayr "cross-validation outcome $R^2_{cv}$ performed best for the coupled method (0.93)..." or something similar.

- The authors are asked to consider adding an overview map locating HRB in China. This could be added to figure 1 or as a fourth sub-figure to figure 7.

**References**

Sauquet, Eric (2006). "Mapping mean annual river discharges: Geostatistical developments for incorporating river network dependencies". In: it Journal of Hydrology 331.1-2, pp. 300-314.

---

## Referee Comment (RC2) · J. O. Skøien (Referee) · 7 Dec 2017

This manuscript describes a coupling of the Budyko approach and hydro-stochastic interpolation. The topic is interesting, the results good, but revisions are necessary before possible publication, particularly related to the presentation.

I am a bit surprised by the relatively poor performance of the application of hydro-stochastic interpolation directly. It is also interesting that two methods that both over-estimate parts of the prediction area can achieve a better result together. I tried to understand how this could be from Figure 7, but the use of different color keys make it difficult to compare the maps. This should be the same for the three maps. I would

also have liked to see the similar map for the observations, and maybe also a map of the residuals. Adding these maps would also help the authors in improving the conclusions, which is currently more like a summary of the results section. I would rather like to see some more discussion around how the combined method can be so much better than the individual methods.

The methodology in Section 2.2 covers almost 5 pages, and is mainly from from Sauquet et al. (2000), somewhat rewritten. It should be shortened, and the text must be more precise.

In Eqs 1-2, is only one w calibrated for all sub-basins, or is it calibrated separately for each sub-basin. If the second, is it then interpolated to uncalibrated locations (or for cross-validation locations)?

The text needs improvement. Copy-editing is necessary, preferably from someone with knowledge about spatial interpolation. A list of necessary edits is given below, but the list is not exhaustive.

Some edits:

P2L14 I think it is better with "relationships between"

P2L19 Maybe rather "spatially interpolate runoff…"

P2L24 determination COefficient?

P2L31 "accurate way in spatial interpolation…" something is wrong.

P3L37 something is missing

P3L43 I think the authors rather want to say that "Geostatistical approaches are commonly used for spatial interpolation".

P3L44-46 "similarity of a generalized stochastic field" – what is meant by this? And what is multivariate here? Rewrite sentence.

P3L47 remove "of values".

P3L49 "kriging is the MOST popular . . ."? (or is A popular)

P3 Kriging -> kriging

P4L57 remove "also suggested as"

P4L63-67 This sentence is not understandable.

P5L87 remove "of"

P94-96 Clumsy sentence.

P6L103-104 I do not understand what is meant here.

P6L111 incorporate -> combine?

P6L114-115 difficult to read, rewrite sentence.

P7L126 what is meant with terrestrial scale here?

P7L138 popularly -> frequently?

P8L152 Delete "interpolation" after Kriging and "The" before "Gottschalk's"

P8L155-L158 The definition of basin area as specific unit should be at L155.

P10L188 (Sauquet et al., 2000) (Sauquet and Gottschalk, 2000) occurs several times, missing the last author.

P14L268 has the highest population density?

P14L272 more than 50% is exploited or water resources are overexploited?

P14L27 "increase difficulty in . . ." -> something seems wrong, revise

P14L279 data packages or digital elevation models?

P17L335 the EMPIRICAL covariance?

P17L342-343 "to obtain the . . .in sub basins A and B" is confusing and can probably be deleted.

P17L350 Maybe "This function is then used for the covariances in the covariance matrix in Eq. (17)."

P17L352 The sentence is clumsy. Also, as MATLAB is mentioned here, I guess it was used for all/most of the analyses? Whether yes or no, it is better to describe in general which software was used, maybe also if there were particular add-on packages.

P18L365 Departures (or deviations) FROM the trend.

P19L380 What is perdition here?

---

## Author Comment (AC1) · 7 Dec 2017

Dear Editor and Reviewer,

We sincerely thank the editor and the reviewer for your reading our previous submission and for your valuable feedbacks that have helped us in improving this manuscript, entitled "*Hydro-stochastic interpolation coupling with Budyko approach for spatial prediction of mean annual runoff*" (ID: hess-2017-472). We have carefully studied the reviewer's constructive comments and made extensive modifications in our revised manuscript. Our point-to-point responses to the reviewer are listed below. The reviewer's comments are laid out in italicized blue font and have been numbered for each comment. Our responses are given in black. In our revised manuscript, all changes are highlighted using blue-colored text. Please contact me if you may have any questions about this revision.

Thank you again for your editorial work that has really helped us a lot.

Sincerely yours,

Xi Chen On behalf of all co-authors

**Major points**

1. 1.63 – 71: This paragraph consists of only two sentences, which are way too long and thus, were confusing for me. In the first sentence the authors make two different points. First, streamflow is a combined landscape information and second, that climatelandscape variability leads to non-stationary runoff observations. I kindly ask the authors to separate these points and reword the following statements in order to foster the structure. Especially the term deterministic term" (l. 65) needs more and clearer introduction. This is in the following work also referred to as deterministic trend"and is of fundamental importance for the proposed method. Introducing this term in more detail will significantly increase my text comprehension for the entire work.

The second sentence in this paragraph (l. 68 - 71) does in my opinion not connect to the first one and it was not obvious what this sentence shall emphasize. What trend does the the spatially nonstationary trend of runoff"(l. 68) refer to? And how is a runoff trend interpretable as hydrological regionalization in terms of hydro-climate and landscape data"(l. 68 - 69)? What I read out of this sentence is that non-stationary runoff is caused by heterogeneity in hydro-climate and the landscape and can be described by empirical relationships as done in the presented studies (l. 71). But this is not exactly what is written down in this paragraph.

In my opinion, the authors shall rewrite the whole paragraph in shorter, non-nested sentences.

Answer: According to the reviewer's suggestion, the paragraph has been rewritten in the revised manuscript (refer to  $l.61 \sim 80$ ). The section of *line.72 \sim 82* in the previous manuscript was moved to *line.104 ~ 111* and *line.114 ~ 117*.

The whole paragraph has been rewritten as:

61 Unlike precipitation or evaporation which we often interpolate to find its values at specific points in space, runoff is an integrated spatial continuous process in basins 62 (Lenton and RodriguezIturbe, 1977; Creutin and Obled, 1982; Tabios and Salas, 1985; 63 Dingman et al., 1988; Barancourt et al., 1992; Bloschl, 2005). Streamflow exhibits some 64 degrees of natural organization or connection of water basins (Dooge, 1986; Sivapalan, 65 2005), e.g., rivers that connect sub-basins. The river network structure constraints water 66 67 balance between upstream and downstream in a basin. The hierarchically organized river structure requires that the sum of interpolated discharge from each of the sub-basins 68 69 equals to the observed runoff in the outlet of the entire basin. Previous studies have indicated that runoff interpolation may overestimate the actual runoff without adequate 70 spatial variation information of runoff (Arnell, 1995), e.g., neglecting the river network 71 72 in connecting sub-basins or processing basin runoff behavior as "points" in space (Villeneuve et al, 1979, Hisdal and Tveito, 1993). Given the obvious nested structure of 73 basins, Gottschalk (1993a and 1993b) developed a hydro-stochastic approach for runoff 74 interpolation. It takes full account of the concept that runoff is an integrated course in 75 the hierarchical structure of river basin systems. Distance between a pair of basins is 76 measured along the river network by geostatistical distance instead of Euclidian distance. 77 The covariogram among points in conventional spatial interpolation is replaced by 78 covariogram between basins. However, in this concept, spatial runoff is considered 79 80 spatially homogeneous over basins (Sauquet, 2006).

The observed patterns of runoff reveal systematic deviations from a purely homogeneous field due to the influence of some deterministic process acting in the basin, such as higher or lower runoff corresponding to larger or smaller rainfall over space. We can describe the hydrological variables of interest in deterministic forms of functions, curves or distributions, and construct conceptual and mathematical models to predict

hydro-climate variability (Wagener et al, 2007). Qiao (1982), Arnell (1992) and Gao et 86 al. (2017) used such methods and derived empirical relationships between runoff and its 87 controlling factors of climate, land use and topography in various basins. However, the 88 deterministic method in describing complex patterns suffers inevitable loss of 89 information (Wagener et al, 2007) because of existence of uncertainty in many 90 hydrological processes and observations. Thus, hydrological variables also contain 91 92 information of stochastic nature and should be treated as outcomes of both deterministic and stochastic processes. Recently, the method of Kriging with an external drift (KED) 93 94 was introduced (Goovaerts, 1997; Li and Heap, 2008; Laaha et al., 2013). It accounts for deterministic patterns of spatial variable and also incorporates the local trend within the 95 neighborhood search window as a linear function of a smoothly varying secondary 96 97 variable, instead of a function of the spatial coordinates.

The inclusion of deterministic terms in the original geostatistical methods has been 98 shown to increase interpolation accuracy of the basin variables, e.g., mean annual runoff 99 (Sauguet, 2006), stream temperature (Laaha et al., 2013) and groundwater table (Holman 100 et al., 2009). Nevertheless, the deterministic term is mostly described by an empirical 101 formula linking the spatial features, e.g. variability of mean annual runoff with elevation 102 (Sauquet, 2006) and relationship between the mean annual stream temperature and 103 altitude of the gauge (Laaha et al., 2013). As a simple semi-empirical approach for 104 105 modelling the deterministic process of runoff, the Budyko framework has been popularly used to analyze relationship between mean annual runoff and climatic factors (e.g., 106 aridity index) over space (Milly, 1994; Koster and Suarez, 1999; Zhang et al., 2001; 107 Donohue et al., 2007; Li et al., 2013; Greve et al., 2014). Many efforts have been devoted 108 to improving the Budyko method by deriving parameters with other external driving 109 factors, such as land use/land cover (Donohue et al., 2007; Li et al., 2013; Han et al., 110

2011; Yang et al., 2007), soil properties (Porporato et al., 2004; Donohue et al., 2012), topography (Shao et al., 2012; Xu et al., 2013; Gao et al., 2017), hydro-climatic variations of seasonality (Milly, 1994; Gentine et al., 2012; Berghuijs et al., 2014) and groundwater levels (Istanbulluoglu et al., 2012). However, it has found that use of the deterministic equation of the Budyko method still came with large errors when it was used in prediction of runoff in any specific basin/area (Potter and Zhang, 2009; Jiang et al., 2015).

2. I would strongly recommend to completely rework the whole section 5 from line

408 to 451, due to many factors. Above all, this whole section is neither a discussion nor

a conclusion in my opinion.

The first paragraph (1.409-424) basically lays the framework for coupling deterministic and statistical models" (1.420), which is used as a justification for the proposed method. The paragraph itself seems to be helpful and relevant but should thus be moved to the introduction, somewhere located (and linked) to the paragraph 1.105-122.

This paragraph is followed by two paragraphs that summarize major parts of the publication. *l.425-434* summarizes the proposed method; while *l.435 - 447* summarizes the reported result.

The only conclusions drawn can be found in the last paragraph (l.448-451). In my opinion, these conclusion are way too general. Furthermore the authors presented a new interpolation method, while long-term climate change impacts are modeled into the future, which would require an extrapolation. Thus, the proposed method is not appropriate to predict climate change impacts.

As the authors presented some interesting results in this publication, it should be easy to draw some more immediate and definite conclusions.

Answer: Discussions and conclusions have been revised in the manuscript. The first paragraph in the previous version of this manuscript ( $l.409 \sim 424$ ) was moved to introduction in the revised manuscript (refer to  $l.64 \sim 66$ ,  $l.83 \sim 86$  and  $l.88 \sim 93$  in the Answer of *Major point 1*). We also revised the conclusions in more details. The last paragraph on long-term climate change impacts was replaced by discussions on reduction of errors from our method.

The section of Discussions and Conclusions from *l*.460~487 in the revised manuscript is listed below.

We have tested our coupled method and compared the results with the Budyko 460 method and the traditional hydro-stochastic interpolation in the Huaihe River basin 461 (HRB) in China. Our results show that the deterministic process controls runoff variation 462 in space instead of the stochastic process over the 40 sub-basins in HRB. The 463 interpolation errors in terms of MAE and RMSE from the Budyko are smaller than those 464 from the hydro-stochastic interpolation, and the cross-validation outcome of the 465 deterministic coefficient  $R_{cv}^2$  from the Budyko method is much larger than that from 466 the traditional hydro-stochastic interpolation. Nevertheless, use of the deviation of runoff 467 from the Budyko method estimation, instead of the observations, as the random error in 468 Gottschalk's interpolation method can further improve interpolation accuracy of spatial 469 runoff. The coupled method outperforms both the Budyko method and the stochastic 470 interpolation by significantly increasing the spatial interpolation and prediction accuracy. 471 The interpolation errors in terms of MAE and RMSE from our coupled method reduce to 472 473 47 and 69mm, respectively, over the 40 sub-basins. The maximum error at HWH is 474 significantly reduced as well. It is 236 mm from our coupled method much smaller than 328 mm from the Budyko method and 448 mm from the hydro-stochastic interpolation. 475 The cross-validation outcome of the deterministic coefficient  $R_{cv}^2$  from our coupled 476 method is 0.93, much larger than 0.81 and 0.54 from the Budyko method and the hydro-477 478 stochastic interpolation, respectively. The prediction from the coupled method captures most accurately the regional high and low runoff in the HRB among the three methods. 479 Our results show that there are still large interpolation errors from our coupled 480 method at some of sub-basins, e.g., in larger sub-basins of ZK and BB where the relative 481 errors are larger than 40%. Such large errors could be resulted from insufficient number 482 of observation stations in such large sub-basins (see Fig. 1). Other possible reasons may 483 be from additionally external factors that drive runoff in space, such as land use/land

484

485 cover, soil properties, topography, hydro-climatic variations of seasonality and

486 groundwater levels. Use of such data in improving Budyko method will enhance our

487 understanding the deterministic process and thus increase the interpolation accuracy.

3. l.136 - 141: To me it is not clear why the authors have chosen Fu's equation. In the introduction to Budyko approaches (l. 129 - 136) the authors introduced a number of adjustments and improvements to the original approach suggested by other studies and highlighted their importance. Fu's equation does not incorporate any of these, but rather a dimensionless model parameter"(l. 144), which does only control the partitioning of precipitation into runoff"(l. 145). The authors are kindly asked to give more insights on this decision. Additionally, the calibration of this parameter is just mentioned in l. 146, but not further described.

Answer: In these adjustments, they only changed the parameter of Budyko equations, e.g., establishing relationship of  $\omega$  in Fu's equation with land surface data. We agree that the improved Budyko approaches in consideration of other driving factors in addition to the aridity index could improve the prediction accuracy of runoff. However, they need plenty of basin characteristics that are often not available or inaccurately quantified in many areas. In our revision, we adopted the parametric Budyko curves of  $E/P \sim E_0/P$ . In the revision, we discussed how this possible way could increase accuracy of spatial runoff interpolation.

The calculation of parameter  $\omega$  is given in *l*.147~154 in the revised manuscript.

The parameter  $\omega$  can be calculated by observed  $P, E_0$  and R at each of gauged 147 sub-basins. The mean value of  $\omega$  can be obtained by averaging  $\omega$  of sub-basins or least 148 square method, i.e. to find the best fitting  $\omega$  in Eqs. (1) or (2) by minimizing the mean 149 absolute error (MAE) between predicted and estimated annual evapotranspiration E (=P-150 R) from the long-term water balance over sub-basins (Legates and McCabe, 1999). With 151 known  $\omega$ , Eq. (2) can be used for prediction of ungauged basin runoff or interpolation 152 of spatial variation of runoff by using meteorological data in the target sub-basins 153 (Parajka and Szolgay, 1998). 154

4. l.240 - 244: For my understanding, this is the key paragraph of the methodology as it describes the actual coupling of Budyko with hydro-stochastic interpolation. I would summarize this as: 1.: Rd(x) in equation (18) is substituted with equation (2) by setting Rd(x) = R. and calculated for all basins. 2.: The residuals between Rd(x) and observed R is calculated for all gauged basins. Further, these residuals are interpolated for all ungauged basins by fesidual kriging"(l.243). and set as Rs(x) 3.: Equation (18) applies as the final result of this study.

Following the cited Hesidual kriging" from Sauquet (2006) it was not clear to me, how exactly the Hesidual kriging is performed on the ungauged basins. The residuals from this study would be described by a first order polynomial (Accounting for spatial heterogenity," last paragraph, in Sauquet (2006)), and be combined with  $\xi q$ , the error in residuals. But, for me, it is not clear how this  $\xi q$  or the g from Sauquet (2006) were calculated. From my point of view, the interpolation scheme described in Sauquet (2006) seems to be closely related to the general approach presented by the authors. Then, the delimitation between the two studies was not clear to me from the introduction. In any case a clarification of how Rs(x) is calculated, how section 2.2 sets in and is linked here would be highly appreciated.

Answer: We completely revised the related paragraphs.  $l.240 \sim 244$  in the previous version of the manuscript was revised as you suggested ( $l.275 \sim 282$ ). In our work, the spatial heterogeneity was described by the Budyko equation, and a deviation from the Budyko method estimation is taken as the residual at all observation stations. Then the original hydro-stochastic interpolation approach was used for interpolation of the residual. The superposition of the estimates of the residual and the Budyko equation yields the prediction of runoff R.

The "residual kriging" is performed on the ungauged basins (e.g., non-overlapping sub-basins) by simultaneously optimizing the weights  $\lambda_j^i$  (i= 1, ..., *M*; j= 1, ..., *n*) according to Eqs. (11)-(15) (see *l.217~235* and *l.275~282* in the revised manuscript).

Our approach is similar to that of Sauquet (2006). A major difference is that we applied a semi-empirical approach of the Budyko, while Sauquet (2006) used an empirical formula (average annual runoff with mean elevation in Fig. 6 of Sauquet (2006)) in his description of spatial heterogeneity over basins (the calculation process of  $R_s(x)$  please refer to *l.404-412* in our revised manuscript).

The content of *l.217~222* beneath Eq.(11) is as follows:

```
where, L_i is the weights column vector with respect to the i - th non-overlapping sub-
```

218 basins, each of which relates to the weights  $\lambda_j^i$  (*i*= 1,..., *M*; *j*=1,..., *n*) of sample

219 observations, and  $\mu^*$  and  $\mu^i$  are all the Lagrange coefficients. To simultaneously

220 optimize the weights  $\lambda_i^i$  (*i*= 1, ..., *M*; *j*= 1, ..., *n*), the covariance function values

221  $Cov(u_i, u_i)$  of pair points in Eqs. (6) and (7) are represented as covariance function

values of the pair sub-basins.

217

The content of *l.229~235* beneath Eqs.(12)-(15) is as follows.

where,  $A_i$  and  $A_j$  are the available basins (*i*=1,2..., *n*; *j*=1,2..., *n*, *n* is the number of gauged sub-basins),  $\Delta A_i$  is the non-overlapping area for sub-basin *i* (*i* = 1, ..., *M*), *Cov*( $A_i, A_j$ ) is theoretical covariance function value of pairs of available basins, *Cov*( $A_n, \Delta A_i$ ) is theoretical covariance function value of an available basin and the predicting basin,  $n_i$  is the number of grids of  $\Delta A_i$ ,  $n_T$  is the number of grids of the whole basin,  $r(A_j)$  is runoff depth for basins *j* (*j* = 1, ..., *n*) with discharge observations,  $V_i^T$  is the transposed column vector of  $V_i$ ,  $\mu^i$  is the Lagrange coefficients.

The content of *l.275~282* is as follows:

In this study, Eq. (2) is used as an external drift function for estimation of the 275 deterministic component  $R_d(x)$  in all basins, i.e.,  $R_d(x)$  in Eq. (18) is substituted for Eq. 276 (2) by setting  $R_d(x) = R$ . The residuals between  $R_d(x)$  and observed R is calculated for all 277 gauged basins. Furthermore, these residuals are interpolated for all ungauged basins by 278 the "residual kriging" (Sauquet, 2006), i.e.,  $R_s(x)$  in Eq. (18) is replaced by  $r^*(A_0)$  in 279 Eq. (3) by setting  $r^*(A_0) = R_s(x)$  for the hydro-stochastic interpolation method in section 280 2.2. The superposition of the estimates of both components according to Eq. (18) yields 281 the prediction of *R*. 282

**The content of *l*.404~412 is as follows:**

The empirical covariogram of  $R_s^*(x)$  for each pair of sub-basins versus sub-basin distances is plotted in Fig. 5. The following exponential function is obtained from best fitting the empirical covariogram

407
$$Cov_p(d) = 3000 \exp(-d/48.34).$$
 (25)

408 From (25), matrices C,  $C_0$ , K, V and G in Eqs. (9)-(15) are calculated using 409 MATLAB, and the weight coefficient matrix of runoff deviation is subsequently 410 calculated to predict runoff deviation. Since this interpolation scheme represents the

- spatial runoff deviation, the sum of the interpolated runoff deviation and the simulated
- runoff by Fu's equation is regarded as the total interpolated runoff in the sub-basins.

5. l.219 - 220: Which scatter diagram are you referring to, here? Furthermore I can hardly imagine how such a diagram would look like. For my understanding, an empirical covariogram relates the separating distances of lag classes the inner-class covariance observed in the data. Please describe how a diagram like this shall be scattered over the distances between all sub-basin combinations. Furthermore, equation (17) presented in line 222 is used to derive a theoretical covariogram. From my understanding (and in fact I am not sure what u1; u2; du1; du2 are referring to here, see minor point below) this will yield a single value Cov(A; B) for sub-basins A and B. Does the theoretical covariogram then relate Cov(A; B) to d(A; B) defined in (16) (l.217)? If so, a more descriptive and clear explanation in the respective paragraph would be highly appreciated. Additionally, do Cov(A; B) (l.220) and Cov(ui; un) (l.178-180) describe the same thing? Answer: We completely revised Section 2.2 for clear description of the empirical and

The content of  $l.245 \sim 264$  is presented below:

- 245 The theoretical covariogram Cov(A, B) is derived in the same way as geostatistical
- distance by averaging the point process covariance function  $Cov_p$ , i.e.,

theoretical covariograms (1.245~264 in the revised manuscript).

247
$$Cov(A,B) = \frac{1}{AB} \int \int_{AB} Cov_p(||u_1 - u_2||) du_1 du_2$$
(17)

where,  $Cov_p(||u_1 - u_2||)$  is the theoretical covariance function value of pairs points

249 with a distance  $||u_1 - u_2||$  in basins A and B.

In the above, the geostatistical distance d(A, B) between *A* and *B* is calculated based on grid division in each of the sub-basin (Sauquet and Gottschalk, 2000). We can obtain the mean distance *d* between all possible pairs of points (center points of grids) within the two sub-basins (*A* and *B*). For *n* sub-basins with observations, there are n(n+1)/2 pairs of the sub-basins, correspondingly with the mean distance  $d_i$  (*i*=1,..., n(n+1)/2).

256 Corresponding to the mean distance  $d_i$  between pairs of the sub-basins, the 257 empirical covariogram  $Cov_e(d_i)$  can be calculated using the runoff depth of pairs of the sub-basins. The geostatistical distances  $d_i$  are then ranged with an interval (50 km in this study) to obtain the mean of  $Cov_e(d_i)$  for the distance interval. Finally, the mean of  $Cov_e(d_i)$  vs. the geostatistical distances  $d_i$  within each of the range intervals is used to draw a scatter diagram of the empirical covariogram  $Cov_e(d_i)$ .

The trial-and-error fitting method is used to calibrate  $Cov_p(d)$  in Eq. (17) aiming to best fit  $Cov_e(d)$ . Only independent sub-basins are used to calculate the covariance function to avoid spatial correlation of the nested sub-basins. 6. The authors should consider to report their result more consistently and comprehensive. Beside a cross-validation, the authors compare the three different interpolation approaches by comparing the errors each method yielded. This error reporting in line 377-379; 355-356 and 328-331 shall be harmonized and report the same numbers.

I would suggest reporting the overall minimum, maximum and mean error found in a single sub-basin, along with the minimum, maximum and mean relative error (as share of basin-specific runoff) found in any sub-basin. Both kind of errors can be reported as a absolute (in mm) and relative (in %) number. In my opinion this makes sense as, for example, the sub-basin yielding the biggest absolute error in equation 2 (which is HWH), does not show the biggest relative error (as eg. SQ shows a bigger relative error). Beside reporting these important numbers, the authors should consider to report these

numbers in table 2, as well.

Answer: We revised discussions of the results from the three methods in a way more consistent to that suggested by the reviewer, and the error reporting in  $l.377 \sim 379$ ; 355 $\sim$ 356 and 328 $\sim$ 331(in the previous manuscript) have been harmonized in same numbers (refer to  $l.413 \sim 415$ ; 390 $\sim$ 392 and 366 $\sim$ 370).

We modified Table 2 by adding those numbers as the reviewer suggested.

The contents of *l*.413~417 is presented below:

- 413 Prediction outcome of runoff is listed in Table 1, with the *MAE* of 47 mm and *RMSE*
- of 69mm over the 40 sub-basins. The largest absolute error is at HWH (236 mm) and the
- smallest at JJJ (2 mm) (Table 2). The largest relative error is about 42.1% of the observed
- 416 runoff at BB station, and the smallest is 0.3% of the observed runoff at JJJ station,
- 417 corresponding to the absolute errors of 90 and 2 mm, respectively.

The contents of 390~394 is presented below:

- 390 The *MAE* and *RMSE* are 134 mm and 176mm, respectively. The largest absolute error is
- at HWH (448 mm) and the smallest at XHD (3 mm) (Table 2). The largest relative error
- is about 85.1% of the observed runoff at ZK station and the smallest is 0.4% of the
- 393 observed runoff at XHD, corresponding to the absolute errors of 105 and 3 mm,
- 394 respectively.

The contents of 366~370 is presented below:

- The *MAE* of Budyko runoff prediction is 94 mm, and the *RMSE* is 112 mm. The largest absolute error is at HWH (328 mm) and the smallest at XX (24 mm) (Table 1 and 2). The largest relative error is about 81.6% of the observed runoff at XZ station and the smallest is 5.0% of the observed runoff at XHD, corresponding to the absolute errors of 91 and 37
- 370 mm at those two stations, respectively.

**The modified Table 2 is as below:**

Table 2 Interpolation cross-validation errors between the predicted and observed runoff at 40 sub-

| basing for the three methods |               |                                |                 |
|------------------------------|---------------|--------------------------------|-----------------|
| Evaluation indicators        | Budyko method | Hydro-stochastic interpolation | Coupling method |
| MAE (mm)                     | 94            | 134                            | 47              |
| MSE (mm 2 )       | 12561         | 31024                          | 4798            |
| RMSE (mm)                    | 112           | 176                            | 69              |
| Max absolute error (mm)      | 328           | 448                            | 236             |
| Min absolute error (mm)      | 24            | 3                              | 2               |
| Max relative error (%)       | 82            | 86                             | 50              |
| Min relative error (%)       | 5             | 0.3                            | 0.3             |
| R 2 cv | 0.81          | 0.54                           | 0.93            |

basins for the three methods

**Minor points**

1. l.322 - 323: This observation is not supported by fig. 3. From my point of view it is not possible to derive the location of a sub-basin from this figure.

Answer: We added locations of the sub-basins in the north and the south section in the revised Figs. 3(b) and 3(c) (also shown below), showing runoff distribution lower in the north and higher in the south.

Figure 3 Plot of (b)  $R \sim E_0/P$  for 40 sub-basins of HRB (c) sub-basins in the north and south of HRB. Note: in Fig 3(b) and (c), blue color indicates wetter climate in south and yellow color indicates the drier climate in north.

2.1.83 - 88: The authors make different points here within one long confusing sentence. They are kindly asked to break this sentence down to the core statements of: 1.: runoff is an integrated spatial continuous process, not a field like precipitation; 2.: runoff interpolation must take the stream network into account; 3.: the stream network constraints the water balance up- and downstream. Furthermore, please clarify the connection between a water balance constraint and assumed runoff properties that can be traced back to field properties.

Answer: Taking the Reviewer's suggestion, we broke this long confusing sentence into several short ones to make it easy to read and understand (refer to  $l.61 \sim 67$  in the Answer of *Major point 1*).

3. l.90 - 91: Please explain l'ateral streamflow'(l. 90). What is that and how is it connected to the topic? None of the two presented studies, that shall explain the link between runoff overestimation and heglecting lateral streamflow" contain the term l'ateral streamflow." Please clarify what the two studies actually indicate.

Answer: We changed this expression as the river network in connecting sub-basins (the content is included in the Answer of *Major point 1*).

4. l.92 – 96. For my understanding, this part is not linked to the other parts of this paragraph or the introduction as far as I read it at that point. Why is this important? Additionally, hydrostochastic interpolation"(l. 92) was not clear to me at that point and the authors might consider some more explanation. Furthermore, the difference between Euclidean distances"(l. 94) used in Conventional stochastic methods"and the Spatial distance"(l. 95) is too vague for me. Consider adding an explanation.

Answer: We deleted this sentence. In the revised manuscript, we added discussion on how to obtain spatial distance between a pair of basins ( $l.238 \sim 244$  and  $l.250 \sim 255$  in the revised manuscript).

The contents of *l.250~255* is presented in the Answer of *Major point 5*.

The contents of  $l.238 \sim 244$  is presented below:

- runoff hierarchical structure in the river system. The appropriate geostatistical distance
- between sub-basins A and B defined by Gottschalk (1993b) is expressed as the
- expectation of distances of all the possible pairs of points inside *A* and *B*:
- 241  $d(A,B) = \frac{1}{AB} \int \int_{AB} ||u_1 u_2|| du_1 du_2$ (16)
- where, A and B are the areas of sub-basins A and B,  $u_1$  and  $u_2$  are the locations of
- pairs of points inside basins A and B,  $du_1$  and  $du_2$  are the differential symbol of  $u_1$
- 244 and  $u_2$ , respectively.

**5. Please clarify what samples refers to in line 171.**

Answer: We revised it as gauged stations in *line 177*. The content of *l.174~178* is presented below:

174

$$\begin{cases} \sum_{j=1}^{n} \lambda_i Cov(u_i, u_j) + \mu = Cov(u_i, u_0), \quad i, j = 1, 2, \dots n\\ \sum_{i=1}^{n} \lambda_i = 1 \end{cases}$$
(4)

where,  $Cov(u_i, u_i)$  is the theoretical covariance function value between each pair of

- gauged stations (i=1,...n, j=1,2...n), and  $Cov(u_i, u_0)$  is the theoretical covariance value
- between the location of interest  $u_0$  and each of the gauged stations  $u_i$ ,  $\mu$  is the
- 178 Lagrange multiplier.

6. l. 362-364: The authors are asked to clarify what trend removal 'refers to here, as no kind of trend removal was reported in the methods. From that, what kind of assumptions do you j'ustify'from applying a trend removal? Do you assume the residuals to be spatially autocorrelated or do you assume an existing spatial autocorrelated random error underlying the residuals themselves, as the hydro-stochastic interpolation'is performed on the residuals?

*Consider extending the corresponding methods part.*

Answer: We deleted this sentence. Here, we assumed the residuals to be spatially auto correlated, which is the basic condition for the stochastic interpolation method.

7. Is the du1; du2 used in (17) (1.222) and (18) (1.236) the same thing, or does the d from (17) refer to the d(A; B) calculated in (16) (1. 217)? If not, what is (16) then used for? If yes, please clarify the difference of the two used du1; du2.
Answer: We clarified the explanations of these items in the revised manuscript (refer to 1.242~244, 1.250~255, and 1.272~274).
The content of 1.242~244 is presented in the Answer of *Minor point* 4.
The content of 1.250~255 is presented in the Answer of *Major point* 5.
The content of 1.272~274 beneath Eq.(18) is presented below:

- 272 In (18), R(x) is runoff at location x,  $R_d(x)$  is the deterministic component of the
- spatial trend and the external drift (Wackernagel, 1995) that results in nonstationary
- 274 variability.

**8. Please describe what spatial variance"(l. 259) exactly means here and how it is defined.**

Answer: Spatial variance is the variance of observed runoff data, and is calculated from this formula:  $V_{NK} = \frac{\sum (R(x_i) - \bar{R})^2}{n-1}$ , where  $\bar{R}$  is mean R(x). We have added this formula in the revised manuscript (*l.296~297*).

**9. The used precipitation data is described to be a Elimatological dataset"(l.287). What kind of data product is this? An interpolated and aggregated map from a observation network? A radar product?**

Answer: The used precipitation data are from the monthly precipitation dataset of China at 0.5° spatial resolution constructed by China Meteorological Administration. The dataset is on the basis of 2472 observational stations of China reorganized by National Meteorological Information Center, using Thin Plate Spline (TPS) interpolation method in ANUSPLIN software to obtain. It offers the monthly precipitation grid data of China started from 1961. The website from where we download the data is:

http://data.cma.cn/data/detail/dataCode/SURF\_CLI\_CHN\_PRE\_MON\_GRID\_0.5.html. (refer to 1.326~328).

**10. 1.290 How was this interpolation conducted? 'ArcGIS''s capable of more than one interpolation method. Please name the method, not the tool.**

Answer: According to the reviewer's advice, this sentence in lines 289 to 290 was modified as:

"Pan evaporation data at 21 meteorological stations in HRB are used to interpolate spatial potential evapotranspiration using ordinary kriging interpolation method via ArcGIS." *(refer to 1.328~330).*

**11. What is the *Helative error* [of] 91 mm? Is this the absolute error at XZ station, where the relative error is the largest observed of 81.6%?**

Answer: It is the absolute error. We have corrected it and pointed out the stations in the revised manuscript *(refer to l.368~370)*.

The content of 1.368~370 is in the Answer of Major points 6.

12. l. 340 - 348: Why was HRB divided into a grid? The corresponding methodological description of these results (l. 212 - 217) did not mention this step. Furthermore, for me the link between equation 24 (l.337), equation 25 (l. 349) and figure 4 is not clear. Both equations describe a empirical covariance C(d), while figure 4 shows a covariance function and with an empirical covariogram." Which one does refer to what here? The authors are kindly asked to make this clearer and the notation more distinct.

Answer: We revised the descriptions in text and Fig. 4. We first calculate empirical covariogram ( $Cov_e(d)$ ) using sub-basin runoff data, and then use it to fit covariance function  $Cov_p(d)$  and obtain theoretical covariogram by the integration of Eq. (17).

This part of modifications in the revised manuscript are shown in  $l.245 \sim 264$  which is listed in the Answer of *Major point 5*.

13. l.351 How shall equation 25 be used to *calculate the theoretical covarinace matrix* Cov(A; B)? In line 220 Cov(A; B) was described as a *covariace matrix* a matrix. Is Covp in equation 17 than the same as C(d) in equation 25? Is the d in equation

25 then derived from equation 16 for each sub-basin pair A,B? Are u1; u2 in equation 16 then the grid points mentioned in 1.340 - 348 or the samples "mentioned in line 171? Clarifying this specific step in the methods wherever appropriate would be highly appreciated.

Answer: We have clarified it on  $l.250 \sim 261$  in the revised manuscript. The content of  $l.250 \sim 261$  is presented in the Answer of *Major point 5*.

14. *l*.403-404 Did you mean that figure 7 (a) and (b) overestimate runoff, instead of underestimate, as stated? Because (a) ranges from 145mm - 280mm in the north and (b) ranges from 140mm - 280mm, in contrast (c) ranges from 60mm to 250mm in the north. Adding another sub-figure to figure 7 showing the measured runoff values can make figure 7 even more meaningful. Additionally, I would strongly recommend using the same value ranges for the color codes in figure 7, this will make the sub-figures more comparable and consistent.

Answer: We revised it according to the reviewer's suggestion (refer to  $l.440 \sim 443$  and Fig. 7(d)).

The revised *Fig.* 7 is shown at the end of this Response letter.

The content of  $l.440 \sim 443$  is presented as follows:

- 440 interpolation methods. Compared with our coupled method (Fig. 7c), the Budyko method
- 441 (Fig. 7a) and hydro-stochastic interpolation (Fig. 7b) markedly overestimate sub-basin
- 442 runoff in the north where climate is relatively dry and runoff is small (ranging from
- 443 140mm 280mm).

**Technical points**

1. In my opinion all the figures should be revised. The figure captions shall be extended and describe all figure elements. This is especially true for figures 3,4,5 and 6. Consider adding legends to figures 3 and 6.

Answer: According to the reviewer's suggestion, we have re-drawn all the figures, revised figure captions, and added legend to Fig. 3(a).

All the revised figures with their captions are shown at the end of this Response letter.

2. The authors are kindly asked to revise all their equations. Please make sure, that all used symbols are explained beneath the equation. This is especially true for  $\mu *$  and  $\mu i$  (l. 189); The sub- or superscripted T used in e.g. in l 192; the undefined symbols u1; u2; du1; du2 (l.217); Cov(ui; un) in l.178-179, 194-197).

Wherever possible the symbol description shall also include the used unit. The unit was only given in a single case.

Answer: Thank you for your kind reminder.

We checked all the equations in the manuscript and made sure all the symbols are given their meanings beneath the equations.

3. 1.129 – 136: This part is in fact a literature review on Budyko approaches and should thus be

**moved from the methodology part into the introduction.**

Answer: We moved them to the Introduction  $(l.108 \sim 114)$  which is shown in the Answer of *Major point 1*.

4. 1.334 - 339: In my opinion, these are methods and should be moved to the correct section. Answer: We described it in the Introduction which is shown in the Answer of Major point 1.

5. *l.314* - *316*: Consider moving this to the methods (*l.147-148*), where the "alibration" is not further described.

Answer: We have already moved the contents of lines  $314 \sim 316$  (in the previous manuscript) to lines  $148 \sim 151$  which is presented in the Answer of *Major point 3*, and explained in more details the "calibration" of parameter  $\omega$ .

6. What exactly is meant by drainage basin'in line 224? In the preceding text the authors referred to basins and sub-basins.

Answer: "drainage basin" in this manuscript is changed to "basins" or "sub-basins" (*l.263* and *l.264*) through this revision.

7. Consider replacing the thod with semi-empirical approach"(l. 112) with the thod with semiempirical Budyko approach," in order to be even more clear here. Answer: We have replaced this term (refer to l. 118 ~119).

8. *l.405* The authors should consider replacing area above BB"with area upstream of BB" area south of BB," to be more precise here.

Answer: We have revised this term in *l.444* in the revised manuscript. We also changed the wording in the caption of Figs. 1 and 2.

9. In line 388, I would not state that 0.93 [is] much larger than 0.81 and 0.54," as 0.93 - 0.81 is in fact smaller than 0.81 - 0.54. I would rather sayr cross-validation outcome  $R^2_{cv}$  performed best for the coupled method (0.93)... "or something similar.

Answer: We changed the expressions to "In terms of the cross-validation outcome in Table 2, our coupled method performed best with  $R_{cv}^2$  as large as 0.93, much larger than 0.81 and 0.54 from the Budyko method and the hydro-stochastic interpolation, respectively" (refer to  $l.424\sim427$ ). It does not refer to the difference value between them (0.93 - 0.81 and 0.81 - 0.54).

10. The authors are asked to consider adding an overview map locating HRB in China. This could be added to figure 1 or as a fourth sub-figure to figure 7

Answer: We have added it in Fig.1. Fig.1 is shown at the end of this Response letter.

**Other modifications:**

Following errors have been corrected in the revision.

**1. Line 141: Equation (2)**

The original equation (2) was:  $R = \left(1 + \left(\frac{E_0}{P}\right)^{\omega}\right)^{\frac{1}{\omega}} - E_0$ , in which the symbol P was

missed and we added it, that is:  $R = P \cdot \left(1 + \left(\frac{E_0}{P}\right)^{\omega}\right)^{\frac{1}{\omega}} - E_0.$

**2. *Line 320: Equation (23)**

The original equation (23) was  $R = \left(1 + \left(\frac{E_0}{P}\right)^{2.213}\right)^{\frac{1}{2.213}} - E_0$ . Similarly, it was modified as  $R = P \cdot \left(1 + \left(\frac{E_0}{P}\right)^{2.213}\right)^{\frac{1}{2.213}} - E_0$ .

**3. Line 149: the words "sub basins"**

"Sub basins" was changed to "sub-basins through the revision.

Other modifications not listed here were highlighted using blue-colored text in the revised manuscript.

All the revised figures are listed below:

---

## Author Response (AR1)

Dear editor and reviewer,

We sincerely thank the editor and the reviewer for your reading our previous submission and your valuable feedbacks have helped us in improving this manuscript. We have carefully studied the reviewer's constructive comments and made extensive modifications in our revised manuscript. The previous title "Hydro-stochastic interpolation coupling with Budyko approach for spatial prediction of mean annual runoff" was changed to "Hydro-stochastic interpolation coupling with Budyko approach for prediction of mean annual runoff". Our point-to-point responses to the reviewer are listed below. The reviewer's comments are quoted in black font and numbered in sequence. Our responses are in italic blue font. In our revised manuscript, all changes are highlighted in blue.

Sincerely yours,

Xi Chen
On behalf of all co-authors

**Major points**

1. l.63 – 71: "This paragraph consists of only two sentences, which are way too long and thus, were confusing for me. In the first sentence the authors make two different points. First, streamflow is a combined landscape information and second, that climate-landscape variability leads to non-stationary runoff observations. I kindly ask the authors to separate these points and reword the following statements in order to foster the structure. Especially the term "deterministic term" (l. 65) needs more and clearer introduction. This is in the following work also referred to as "deterministic trend" and is of fundamental importance for the proposed method.

Introducing this term in more detail will significantly increase my text comprehension for the entire work.

The second sentence in this paragraph (l. 68 – 71) does in my opinion not connect to the first one and it was not obvious what this sentence shall emphasize. What trend does the "the spatially nonstationary trend of runoff" (l. 68) refer to? And how is a runoff trend interpretable as "hydrological regionalization in terms of hydro-climate and landscape data" (l. 68 – 69)? What I read out of this sentence is that non-stationary runoff is caused by heterogeneity in hydro-climate and the landscape and can be described by empirical relationships as done in the presented studies (l. 71). But this is not exactly what is written down in this paragraph.

In my opinion, the authors shall rewrite the whole paragraph in shorter, non-nested sentences."

*Answer: Complying with the reviewer's suggestions, the paragraph has been rewritten in the revised manuscript (l.61-80). The "deterministic term" is described in more detail (l.81-90).*

*The second sentence in the paragraph (l. 68 – 71) has also been revised.*

*The section from lines 72-82 in the previous manuscript was moved to lines 103-115.*

2. "I would strongly recommend to completely rework the whole section 5 from line 408 to 451, due to many factors. Above all, this whole section is neither a discussion nor a conclusion in my opinion.

The first paragraph (l.409-424) basically lays the framework for coupling "deterministic and statistical models" (l.420), which is used as a justification for the proposed method. The paragraph itself seems to be helpful and relevant but should thus be moved to the introduction, somewhere located (and linked) to the paragraph l.105-122.

This paragraph is followed by two paragraphs that summarize major parts of the publication. l.425-434 summarizes the proposed method; while l.435 - 447 summarizes the reported result.

The only conclusions drawn can be found in the last paragraph (l.448-451). In my opinion, these conclusion are way too general. Furthermore the authors presented a new interpolation method, while long-term climate change impacts are modeled into the future, which would require an extrapolation. Thus, the proposed method is not appropriate to predict climate change impacts.

As the authors presented some interesting results in this publication, it should be easy to draw some more immediate and definite conclusions."

*Answer: Discussions and conclusions have been revised in the revision. The first paragraph in the previous version of this manuscript (lines 409-424) was moved to introduction in the revised manuscript. We also revised the conclusions with more conclusive statements.*

*The last paragraph on long-term climate change impacts was replaced by discussions on improved accuracy of predicted runoff from our coupled method.*

3. l.136 – 141: "To me it is not clear why the authors have chosen Fu's equation. In the introduction to Budyko approaches (l. 129 – 136) the authors introduced a number of adjustments and improvements to the original approach suggested by other studies and highlighted their importance. Fu's equation does not incorporate any of these, but rather a "dimensionless model parameter" (l. 144), which does only control the "partitioning of precipitation into runoff" (l. 145). The authors are kindly asked to give more insights on this decision. Additionally, the calibration of this parameter is just mentioned in l. 146, but not further described."

*Answer: In those adjustments of Budyko approaches, Fu's equation has been used but the parameter of Budyko equation is further quantified by establishing relationship of ω with land surface data. We agree that the improved Budyko approaches in consideration of other driving factors in addition to the aridity index could improve the prediction accuracy of runoff. However, they need many basin characteristics that are often unavailable or inaccurately measured in limited locations. Our study demonstrates how the deterministic term from Fu's equation can help improve the spatial interpolation. In the revised manuscript, we add discussions on these specifics to help clarify our approach. The calculation procedures of parameter ω are described in lines 147-153 and 312-320 in the revised manuscript.*

4. l.240 - 244: "For my understanding, this is the key paragraph of the methodology as it describes the actual coupling of Budyko with hydro-stochastic interpolation. I would summarize this as: 1.: Rd(x) in equation (18) is substituted with equation (2) by setting Rd(x) = R. and calculated for all basins. 2.: The residuals between Rd(x) and observed R is calculated for all gauged basins. Further, these residuals are interpolated for all ungauged basins by "residual kriging" (l.243). and set as Rs(x) 3.: Equation (18) applies as the final result of this study.

Following the cited "residual kriging" from Sauquet (2006) it was not clear to me, how exactly the "residual kriging" is performed on the ungauged basins. The residuals from this study would be described by a first order polynomial ("Accounting for spatial heterogenity", last paragraph, in Sauquet (2006)), and be combined with $\xi q$, the error in residuals. But, for me, it is not clear how this $\xi q$ or the g from Sauquet (2006) were calculated. From my point of view, the interpolation scheme described in Sauquet (2006) seems to be closely related to the general approach presented by the authors. Then, the delimitation between the two studies was not clear to me from the introduction. In any case a clarification of how Rs(x) is calculated, how section 2.2 sets in and is linked here would be highly appreciated."

*Answer: We completely revised the referred paragraphs. Descriptions in lines 240-244 in the previous version of the manuscript were revised according to your suggestion (lines 241-249 in the revised manuscript).*

*In our work, a deviation from the estimation using Budyko method is taken as the residual at all observation stations. Then the hydro-stochastic interpolation approach was used to interpolate the residual. The superposition of these estimates yields the prediction of runoff R.*

*The "residual kriging" is performed in the ungauged basins (non-overlapping sub-basins) by simultaneously optimizing the weights $\lambda_j^i$ (i= 1, ..., M; j= 1, ..., n) (see lines 179-201 and 244-249 in the revised manuscript).*

*Our coupling approach is similar to that of Sauquet (2006). A major difference of our approach from that of Sauquet (2006) is that we applied a semi-empirical approach of*

*the Budyko, while Sauquet (2006) used an empirical formula (average annual runoff with mean elevation in Fig. 6 of Sauquet (2006)) in his description of spatial heterogeneity over basins (see lines 97-103 in revised Introduction).*

*The calculation procedure of Rs(x) is described in lines 244-249 and 366-369 in our revised manuscript.*

5. l.219 - 220: "Which scatter diagram are you referring to, here? Furthermore I can hardly imagine how such a diagram would look like. For my understanding, an empirical covariogram relates the separating distances of lag classes the inner-class covariance observed in the data. Please describe how a diagram like this shall be scattered over the distances between all sub-basin combinations. Furthermore, equation (17) presented in line 222 is used to derive a theoretical covariogram. From my understanding (and in fact I am not sure what u1; u2; du1; du2 are referring to here, see minor point below) this will yield a single value Cov(A; B) for sub-basins A and B. Does the theoretical covariogram then relate Cov(A; B) to d(A; B) defined in (16) (l.217)? If so, a more descriptive and clear explanation in the respective paragraph would be highly appreciated.

   Additionally, do Cov(A; B) (l.220) and Cov(ui; un) (l.178-180) describe the same thing?"

*Answer: We completely revised Section 2.2 to describe more clearly the empirical and theoretical covariogram (lines 202-229).*

6. "The authors should consider to report their result more consistently and comprehensive. Beside a cross-validation, the authors compare the three different interpolation approaches by comparing the errors each method yielded. This error reporting in line 377-379; 355-356 and 328-331 shall be harmonized and report the same numbers.

   I would suggest reporting the overall minimum, maximum and mean error found in a single sub-basin, along with the minimum, maximum and mean relative error (as share of basin-specific runoff) found in any sub-basin. Both kind of errors can be reported as an absolute (in mm) and relative (in %) number. In my opinion this makes sense as, for example, the sub-basin yielding the biggest absolute error in equation 2

(which is HWH), does not show the biggest relative error (as eg. SQ shows a bigger relative error).

Beside reporting these important numbers, the authors should consider to report these numbers in table 2, as well."

*Answer: We revised descriptions of the results from the three methods in a way more consistent to that suggested by this reviewer. Our discussions on prediction error in lines 328-331, 355-356, and 377-379 in the previous manuscript have been revised to make them more coherent (lines 330~336; 355~363 and 378~382 in revised manuscript).*
*We revised Table 2 by adding those numbers as suggested by this reviewer.*

**Minor points**

1. l.322 - 323: "This observation is not supported by fig. 3. From my point of view it is not possible to derive the location of a sub-basin from this figure."

*Answer: We added locations of the sub-basins in revised Figs. 3b and 3c, and also showed lower runoff in the north and higher runoff in the south sub-basins.*

2. l.83 – 88: "The authors make different points here within one long confusing sentence. They are kindly asked to break this sentence down to the core statements of: 1.: runoff is an integrated spatial continuous process, not a field like precipitation; 2.: runoff interpolation must take the stream network into account; 3.: the stream network constraints the water balance up- and downstream. Furthermore, please clarify the connection between a water balance constraint and assumed runoff properties that can be traced back to field properties."

*Answer: Complying with the reviewer's suggestion, we broke this long confusing sentence into several short ones to make it easier to read and understand (lines 61-67).*

3. l.90 – 91: "Please explain "lateral streamflow" (l. 90). What is that and how is it connected to the topic? None of the two presented studies, that shall explain the link between runoff overestimation and "neglecting lateral streamflow" contain the term "lateral streamflow". Please clarify what the two studies actually indicate."

*Answer: We changed this expression as the river network in connecting sub-basins.*

4. l.92 – 96. "For my understanding, this part is not linked to the other parts of this paragraph or the introduction as far as I read it at that point. Why is this important? Additionally, "hydro-stochastic interpolation" (l. 92) was not clear to me at that point and the authors might consider some more explanation. Furthermore, the difference between "Euclidean distances" (l. 94) used in "conventional stochastic methods" and the "spatial distance" (l. 95) is too vague for me. Consider adding an explanation."

*Answer: We deleted this sentence. In the revised manuscript, we added descriptions on how to obtain spatial distance between a pair of sub-basins (lines 204-210 and 216-220).*

5. "Please clarify what "samples" refers to in line 171."

*Answer: We revised it as gauged stations in line 176.*

6. l. 362-364: "The authors are asked to clarify what "trend removal" refers to here, as no kind of trend removal was reported in the methods. From that, what kind of assumptions do you "justify" from applying a trend removal? Do you assume the residuals to be spatially autocorrelated or do you assume an existing spatial autocorrelated random error underlying the residuals themselves, as the "hydro-stochastic interpolation" is performed on the residuals?
Consider extending the corresponding methods part."

*Answer: We deleted this sentence. Here, we assumed the residuals to be spatially auto-correlated, which is the basic condition for the stochastic interpolation method.*

7. "Is the du1; du2 used in (17) (l.222) and (18) (l.236) the same thing, or does the d from (17) refer to the d(A; B) calculated in (16) (l. 217)? If not, what is (16) then used for? If yes, please clarify the difference of the two used du1; du2."

*Answer: We clarified the explanations of these items in the revised manuscript (lines 208-229 and 238-240).*

8. "Please describe what "spatial variance" (l. 259) exactly means here and how it is defined."

*Answer: Spatial variance is the variance of observed runoff data, and is calculated from:$V_{NK} = \frac{\sum (R(x_i) - \bar{R})^2}{n-1}$, in which $\bar{R}$ is mean $R(x)$. We have added this formula in the revised manuscript (lines 264-265).*

9. "The used precipitation data is described to be a "climatological dataset" (l.287). What kind of data product is this? An interpolated and aggregated map from a observation network? A radar product?"

*Answer: The precipitation data are from the monthly precipitation dataset of China with 0.5-degree spatial resolution. The dataset was developed by China Meteorological Administration, based on 2472 observational stations in China. It contains gridded monthly precipitation data of China started from 1961. The website from where we download these data is:*

*http://data.cma.cn/data/detail/dataCode/SURF_CLI_CHN_PRE_MON_GRID_0.5.html. This information is added in the revision in lines 290-295.*

10. "l.290 How was this interpolation conducted? "ArcGIS" is capable of more than one interpolation method. Please name the method, not the tool."

*Answer: It was revised to:*

*"Pan evaporation data at 21 meteorological stations in HRB are used to interpolate $E_0$ using the ordinary kriging interpolation method and ArcGIS." (lines 295-297 in the revised manuscript).*

11. "What is the "relative error [of] 91 mm"? Is this the absolute error at XZ station, where the relative error is the largest observed of 81.6%?"

*Answer: It is the absolute error. We have revised it and also given the sub-basin's name where those errors are observed.*

12. "l. 340 - 348: Why was HRB divided into a grid? The corresponding methodological description of these results (l. 212 - 217) did not mention this step. Furthermore, for me the link between equation 24 (l.337), equation 25 (l. 349) and figure 4 is not clear. Both equations describe a empirical covariance C(d), while figure 4 shows a "covarinance function" along with an "empirical covariogram". Which one does refer to what here? The authors are kindly asked to make this clearer and the notation more distinct."

*Answer: We described the grid-based estimation of runoff and distance in the revised methodology (lines 194-197 and 216-220 in the revised manuscript).*

*We also revised the descriptions in text and Fig. 4. We first calculate empirical*

*covariogram ($Cov_e$ (d)) using sub-basin runoff data, and then use it to fit covariance*

*function $Cov_p$ (d) for obtaining theoretical covariogram by the integration of Eq. (8)*

*(lines 221-229 in the revised manuscript).*

13. "l.351 How shall equation 25 be used to "calculate the theoretical covarinace matrix Cov(A; B)"? In line 220 Cov(A; B) was described as a "theoretical covariogram", not a matrix. Is Covp in equation 17 than the same as C(d) in equation 25? Is the d in equation 25 then derived from equation 16 for each sub-basin pair A,B? Are u1; u2 in equation 16 then the grid points mentioned in l.340 - 348 or the "samples" mentioned in line 171? Clarifying this specific step in the methods wherever appropriate would be highly appreciated."

*Answer: We have clarified the descriptions on lines 216 -229 in the revised manuscript.*

14. "l.403-404 Did you mean that figure 7 (a) and (b) overestimate runoff, instead of underestimate, as stated? Because (a) ranges from 145mm - 280mm in the north and (b) ranges from 140mm - 280mm, in contrast (c) ranges from 60mm to 250mm in the north. Adding another sub-figure to figure 7 showing the measured runoff values can make figure 7 even more meaningful. Additionally, I would strongly recommend using the same value ranges for the color codes in figure 7, this will make the sub-figures more comparable and consistent."

*Answer: We revised it according to the reviewer's suggestions (lines 402-411 and revised Fig.7(d)).*

**Technical points**

1. "In my opinion all the figures should be revised. The figure captions shall be extended and describe all figure elements. This is especially true for figures 3,4,5 and 6. Consider adding legends to figures 3 and 6."

*Answer: According to the reviewer's suggestion, we have re-drawn all the figures, revised figure captions, and added legend to Figs. 3 and 6.*

2. "The authors are kindly asked to revise all their equations. Please make sure, that all used symbols are explained beneath the equation. This is especially true for $\mu*$ and

μi (l. 189); The sub- or superscripted T used in e.g. in l 192; the undefined symbols u1; u2; du1; du2 (l.217); Cov(ui; un) in l.178-179, 194-197) .

Wherever possible the symbol description shall also include the used unit. The unit was only given in a single case."

*Answer: Thank you for your kind reminder.*

*We checked all the equations in the manuscript and made sure all the symbols are given their meanings after the equations.*

3. "l.129 – 136: This part is in fact a literature review on Budyko approaches and should thus be moved from the methodology part into the introduction."

*Answer: We have moved them to the Introduction.*

4. "l.334 - 339: In my opinion, these are methods and should be moved to the correct section."

*Answer: We described them in the Introduction.*

5. "l.314 - 316: Consider moving this to the methods (l.147-148), where the "calibration" is not further described."

*Answer: We moved the contents of lines 314-316 in the previous manuscript to lines 147-153 in the revised manuscript, and explained in more details the "calibration" of parameter $\omega$*

6. "What exactly is meant by "drainage basin" in line 224? In the preceding text the authors referred to basins and sub-basins."

*Answer: The phrase "drainage basin" in this manuscript has been changed to "basins" or "sub-basins."*

7. "Consider replacing "method with semi-empirical approach" (l. 112) with "method with semi-empirical Budyko approach", in order to be even more clear here."

*Answer: Replaced.*

8. "l.405 The authors should consider replacing "area above BB" with "area upstream of BB" or "area south of BB", to be more precise here."

*Answer: We have revised this term. We also changed the wording in the revised manuscript.*

9. "In line 388, I would not state that "0.93 [is] much larger than 0.81 and 0.54", as 0.93 - 0.81 is in fact smaller than 0.81 - 0.54. I would rather say "cross-validation outcome Rcv2 performed best for the coupled method (0.93)..." or something similar."

*Answer: We changed the expressions to "In cross-validation (Table 2), our coupled method has $R_{cv}{}^2=0.93$, much larger than 0.81 and 0.54 from the Budyko method and the hydro-stochastic interpolation, respectively" (lines 391-393).*

10. "The authors are asked to consider adding an overview map locating HRB in China. This could be added to figure 1 or as a fourth sub-figure to figure 7"

*Answer: We have added it in Fig.1.*

**Other modifications:**

*All the following errors have been corrected in the revision.*

1. Line 141: Equation (2)

*The original equation (2) was: $R = \left(1 + \left(\frac{E_0}{P}\right)^\omega\right)^{\frac{1}{\omega}} - E_0$, in which the symbol P was missed. It has been added, and it is now $R = P \cdot \left(1 + \left(\frac{E_0}{P}\right)^\omega\right)^{\frac{1}{\omega}} - E_0$.*

2. Line 320: Equation (23)

*The revised Equation (14) was written $R = \left(1 + \left(\frac{E_0}{P}\right)^{2.213}\right)^{\frac{1}{2.213}} - E_0$, again missing P in the first term. The missing P has been added.*

3. Line 149: the words "sub basins"

*"Sub basins" was changed to "sub-basins" through the revision.*

**Response to RC2 by J. O. Skøien (Referee)**

*We sincerely thank the reviewer for carefully reviewing our manuscript and for thoughtful feedbacks. We have revised our manuscript taking into account every suggestion and comment of this reviewer. Our point-to-point responses are detailed below.*

1. "This manuscript describes a coupling of the Budyko approach and hydro-stochastic interpolation. The topic is interesting, the results good, but revisions are necessary before possible publication, particularly related to the presentation."

*Answer: We thank Dr. Skøien's suggestions. We have thoroughly revised the manuscript.*

2. "I am a bit surprised by the relatively poor performance of the application of hydro-stochastic interpolation directly. It is also interesting that two methods that both over-estimate parts of the prediction area can achieve a better result together. I tried to understand how this could be from Figure 7, but the use of different color keys makes it difficult to compare the maps. This should be the same for the three maps. I would also have liked to see the similar map for the observations, and maybe also a map of the residuals. Adding these maps would also help the authors in improving the conclusions, which is currently more like a summary of the results section. I would rather like to see some more discussion around how the combined method can be so much better than the individual methods."

*Answer: We remade the four figures using the same color code. We also added Fig. 7d to show the observations as required by this reviewer. The improved figures clearly show spatial differences of estimated runoff by the three methods. The improved statistical results of the residuals are shown in Table 2. We also revised the conclusions and discussed additional advantages of our coupled method and its future improvements.*

3. "The methodology in Section 2.2 covers almost 5 pages, and is mainly from Sauquet et al. (2000), somewhat rewritten. It should be shortened, and the text must be more precise."

*Answer: Complying with the reviewer's suggestion, we have revised Section 2.2 and removed the weight matrix calculation equations which can be found in Sauquet et al.*

*(2000). We also revised some descriptions on water balance constraints, the theoretical covariance function and geostatistical distance.*

4. "In Eqs 1-2, is only one ω calibrated for all sub-basins, or is it calibrated separately for each sub-basin. If the second, is it then interpolated to uncalibrated locations (or for cross-validation locations)?"

*Answer: We used the mean of ω in interpolation. The mean of ω in Eqs. 1 and 2 is calculated at each sub-basin (Table 1), and the ω values at the sub-basins are averaged. In validation, the mean ω is alternatively obtained by fitting the curve of Eq. 1, i.e., E/P~E0/P (E =P-R), from observations to minimize the mean absolute error (MAE) (refer to P8L147-153 and 312-320 in the revised manuscript).*

5. "The text needs improvement. Copy-editing is necessary, preferably from someone with knowledge about spatial interpolation. A list of necessary edits is given below, but the list is not exhaustive."

*Answer: We have carefully improved the grammar of the text. Our responses to the suggested necessary edits are listed below, and corrections are highlighted in blue in our revised text.*

Some edits:

P2L14 I think it is better with "relationships between"

*Answer: It has been changed to "relationships between".*

P2L19 Maybe rather "spatially interpolate runoff: : :"

*Answer: It has been changed to "spatially interpolate runoff in…".*

P2L24 determination Coefficient?

*Answer: "The coefficient of determination" has been changed to "The determination coefficient".*

P2L31 "accurate way in spatial interpolation: : :" something is wrong.

*Answer: This sentence has been modified as "…offers an effective and accurate way to predict mean annual runoff in river basins"*

P3L37 something is missing

*Answer: This sentence has been revised as "Runoff observed at the outlet of a basin is a crucial element for investigating the hydrological cycle of the basin. The runoff is influenced by both deterministic and stochastic processes" (refer to P3L36-38).*

P3L43 I think the authors rather want to say that "Geostatistical approaches are commonly used for spatial interpolation".

*Answer: This sentence is used.*

P3L44-46 "similarity of a generalized stochastic field" – what is meant by this? And what is multivariate here? Rewrite sentence.

*Answer: The sentence has been rewritten (P3L45-48 in the revised manuscript).*

P3L47 remove "of values".

*Answer: It has been deleted in the revised manuscript.*

P3L49 "kriging is the MOST popular: : :"? (or is A popular)

*Answer: This sentence has been rewritten as "The values obtained by geostatistical or kriging interpolation methods are the best linear unbiased estimate…" (refer to P3L50-52 in the revised manuscript).*

P3 Kriging -> kriging

*Answer: It has been changed to "kriging" in the revised manuscript.*

P4L57 remove "also suggested as"

*Answer: The Introduction has been revised thoroughly. This expression has been deleted in the revised section.*

P4L63-67 This sentence is not understandable.

*Answer: This sentence has been rewritten in P5L81-85 in the revised manuscript.*

P5L87 remove "of"

*Answer: Removed.*

P94-96 Clumsy sentence.

*Answer: This paragraph has been rewritten in P4L76-77.*

P6L103-104 I do not understand what is meant here.

*Answer: The paragraph including this sentence has been rewritten (P5L79-80 in the revised manuscript).*

P6L111 incorporate -> combine?

*Answer: The word has been changed to "combine" (refer to P6L116).*

P6L114-115 difficult to read, rewrite sentence.

*Answer: The sentence has been rewritten as "In this study, the spatial runoff from sub-basins in the HRB is separated into the deterministic trend and its residuals both of which are estimated by the Budyko framework and interpolation method." (P6L118-120).*

P7L126 what is meant with terrestrial scale here?

*Answer: It has been changed as "a regional or global scale".*

P7L138 popularly -> frequently?

*Answer: It has been changed to "frequently".*

P8L152 Delete "interpolation" after Kriging and "The" before "Gottschalk's"

*Answer: Corrected (refer to P8L157).*

P8L155-L158 The definition of basin area as specific unit should be at L155.

*Answer: The definition of basin area as a specific unit has been moved to P8L161 in the revised manuscript.*

P10L188 (Sauquet et al., 2000) (Sauquet and Gottschalk, 2000) occurs several times, missing the last author.

*Answer: We have corrected this citation in the revised manuscript.*

P14L268 has the highest population density?

*Answer: The word has been changed to "highest" (refer to P13L274).*

P14L272 more than 50% is exploited or water resources are overexploited?

*Answer: The sentence has been revised to "More than 50% of the water resources is exploited" (refer to P14L277 in the revised manuscript).*

P14L276 "increase difficulty in : : :" -> something seems wrong, revise

*Answer: The sentence has been deleted.*

P14L279 data packages or digital elevation models?

*Answer: The river system shapefiles in ArcGIS are included in the data package from the National Fundamental Geographic Information System issued by National Geomatics Center of China.*

P17L335 the EMPIRICAL covariance?

*Answer: It is correct. We described it in more detail in the Methodology section.*

P17L342-343 "to obtain the : : :in sub basins A and B" is confusing and can probably be deleted.

*Answer: We revised this sentence to make it clearer (P16L341- P17L344 in the revised manuscript).*

P17L350 Maybe "This function is then used for the covariances in the covariance matrix in Eq. (17)."

*Answer: The sentence is rewritten as "This function is further used in calculation of the average theoretical covariances Cov(A,B) in Eq. (8)"(refer to P17L352-353 in the revised manuscript).*

P17L352 The sentence is clumsy. Also, as MATLAB is mentioned here, I guess it was used for all/most of the analyses? Whether yes or no, it is better to describe in general which software was used, maybe also if there were particular add-on packages.

*Answer: We revised this sentence as "Subsequently, the weight matrices are determined using our program in MATLAB" (P17L353-L354).*

P18L365 Departures (or deviations) FROM the trend.

*Answer: It has been changed to "the deviations from the trend…".*

P19L380 What is perdition here?

*Answer: The word should be "prediction". It has been corrected.*

**The list of all relevant changes in the revised manuscript**

1. *P1L1:* The title of the manuscript (we have deleted the word of "spatial");

2. *P2L15:* It has been changed to "between the runoff…";

3. *P2L20:* It has been changed to "spatially interpolate runoff…";

4. *P2L25:* It has been changed to "The determination coefficient for…";

5. *P2L29-30:* It has been changed to "the coupled method offers an effective and accurate way to predict mean annual runoff in river basins.";

6. *P2L32-33:* Some of the Keywords of the manuscript have been changed;

7. *P3L36-38:* The first sentence of Introduction has been revised;

8. *P3L42-45:* This sentence has been revised;

9. *P3L47:* It has been revised as "…referring to more than one variable";

10. *P3L50-51:* The sentence has been changed;

11. *P3L52:* It has been changed to "ordinary kriging";

12. *P4L61-69:* These sentences have been changed;

13. *P4L71-72:* It has been changed to "the river network in connecting sub-basins";

14. *P5L80:* It has been changed to "the expected value of runoff is a constant in space";

15. *P5L81-96:* The paragraph has been rewritten;

16. *P6L103-115:* These sentences have been revised;

17. *P6L116:* It has been changed to "combine";

18. *P6L119-120:* It has been changed to "…both of which are estimated by the Budyko framework and interpolation method";

19. *P6L120- P7L128:* Some of the words have been revised;

20. *P7L130:* It has been changed to "Methodologies";

21. *P7L133:* It has been changed to "regional or global scale";

22. *P7L139:* It has been changed to "used frequently";

23. *P7L142:* Eq. (2) has been revised;

24. *P8L147-153:* The paragraph has been rewritten;

25. *P8L157:* It has been changed to "kriging method" and "Gottschalk's method";

26. *P8L158:* It has been changed to "…basins and identifies the river network and supplemental…";

27. *P8L161:* It has been changed to "…a specific unit of an area $A_0$ in a basin…";

28. *P9L167-172:* Some expressions have been revised;

29. *P9L173:* Eq. (4) has been revised;

30. *P9L176:* It has been changed to "the gauged stations";

31. *P9L179-183:* The paragraph has been rewritten;

32. *P9L186-187:* The sentence has been changed;

33. *P10L194-200:* The paragraphs have been rewritten;

34. *P10L208-210:* The sentences have been revised;

35. *P11L214-229:* These paragraphs have been rewritten;

36. *P12L234-235:* This sentence has been rewritten;

37. *P12L241-249:* The paragraph has been rewritten;

38. *P13L264-265:* The sentence has been revised;

39. *P13L272-273:* The sentence has been revised;

40. *P13L274:* It has been changed to "highest";

41. *P14L277:* It has been changed to "more than 50% of the water resources is exploited";

42. *P14L280-282:* The sentence has been rewritten;

43. *P14L294-297:* These sentences have been rewritten;

44. *P15L314-318:* These sentences have been revised;

45. *P16L322:* Eq. (14) has been revised;

46. *P16L324-329:* The paragraph has been revised;

47. *P16L330-336:* The descriptions of the results have been revised in a way more consistent and coherent;

48. *P16L342- P17L344:* The sentence has been revised;

49. *P17L349- 350:* The sentence has been changed;

50. *P17L352- 354:* These sentences have been rewritten;

51. *P17L355-363:* The descriptions of the results have been revised;

52. *P18L374-375:* The sentence has been revised;

53. *P18L378-382:* The descriptions of the results have been revised;

54. *P18L384:* It has been changed to "Comparisons of predicted runoff by the three methods";

55. *P19L391-393:* The sentence has been revised;

56. *P19L403-406:* The sentence has been revised;

57. *P20L410-411:* The sentence has been revised;

58. *P20L424- P21L433:* The paragraph has been rewritten;

59. *P21L446- P22L456:* The paragraph has been rewritten;

60. *P30L679-681:* Table 2 has been revised;

61. *P27L657- P22L672:* The captions of all figures have been revised;

62. *P31L683- P35L720:* All the figures have been redrawn.

[revised manuscript text omitted]

---

## Referee Report (RR1)

**Answer to Xi Chen's response on my referee comment RC1**

First, I would like to thank the authors for the revised manuscript. Overall, all major and minor points made in my first comment were tackled by the authors. Major parts of the manuscript were revised and the majority of the figures were re-drawn. The manuscript is way more complete at the current stage and the structure was fundamentally improved. Thus, the manuscript can be accepted from my point of view.
Best,

Mirko Mälicke

Major Points :

1 and 2: The authors accepted the reviewer suggestions and revised / rewrote major parts of the mentioned sections. These became much clearer now.

3. The decision for using Fu's equation was clarified, as well as some theoretical background was added. This improvement was really important and helped be understanding the method right away.

4. The authors completely revised the mentioned paragraphs here. The chosen methods are described in more detail and their originating publications were set to scene more precisely, for my understanding.

5 and 6 The authors accepted the reviewers suggestions and revised the affected paragraphs. The result presentation is much more complete now, to my understanding.

Minor Points:

The minor points were all incorporated as suggested by the reviewer. Especially the revision of the methods part fostered a deeper understanding.

Technical Points:

The technical points were all incorporated as suggested by the reviewer. On top the authors made corrections to two equations and added missing symbols.

The figures were revised, but special care should be taken of figure Figure 6 again. Here, the axis labels and tickmarks overlay and are thus not readable everywhere. The authors could used are shared y-axis and omit redundant tickmarks and labels where possible.

---

## Author Response (AR2)

Dear Drs. Skøien and Zehe,

We thank you for your very careful evaluation of every detail in our work. Your evaluations and suggestions have helped us greatly in not only improving the clarity of this work but also the very correctness of our results. Following your recent suggestions, we have re-examined our calculation routines/programs and found two mistakes in our Matlab routines used in the hydro-interpolation. We have corrected those errors and redid all calculations. Our new results are more consistent and in-line with what you have expected. They provide stronger support to our previous conclusions that the coupled method gives better interpolations of the runoff. These improved results have been included in our revised manuscript. The details of how they are included in the revision are described in the attached point-to-point explanation.

Sincerely yours,

Xi Chen
On behalf of all co-authors

Reply

Comment: "*Now that you have included the observations in Figure 7, I am still very puzzled by the poor results of the hydro-stochastic interpolation alone (Fig 7b). Having worked quite a bit with this type of interpolation myself, I think the method should be able to achieve better results than what can be seen in this figure, and that the difference between the methods might be too large.*"

Reply: We re-examined our analyses and found a mistake in our Matlab program for the hydro-interpolation computation. Being specific, in making predictions and cross-validations at a target station we mismatched the weights of neighboring stations. After correcting this mistake, we made recalculations. In the revised results, the error of predicted hydro-stochastic runoff is considerably reduced. The determination coefficient of the cross validation is now 0.71, much higher than 0.58 in the previous result. We also found that we misused the positive and negative signs of the residence values (observed runoff minus the Budyko prediction) in our Matlab program for our combined method. That mistake has also been corrected. The corrected determination coefficient of the cross validation in our combined method is 0.87 (lower than 0.93 in previous result) (see Table 1 in the revised manuscript). Except for these two programming mistakes, we found our calculations are correct.

Comment: "*I still think it should be possible to achieve better results with your combined method, but when both methods appear as mediocre (as in Fig 7a and 7b) then it is surprising to see an improvement as in Fig 7c, particularly as the two methods seem to overestimate and underestimate at the same locations. I think the interpolation worked better for the residuals, but it would be good to know if this is because the residuals are actually easier to interpolate, or if this is because the interpolation of raw data failed for some other reason. If it is the first case, this should be better explained in the manuscript, as it would point out a weakness with these type of interpolation methods. If it is the second case, then the interpolation of raw data would get a better result, but still most likely not as good as the combined method, and most of the manuscript can be left as it is.*"

Reply: After we corrected the calculation routines, overestimated and underestimated runoff from the Budyko and the hydro-stochastic interpolation are not happening at the same locations (shown in revised Figs. 7a and 7b). The interpolation for the residuals works no better than the interpolation using the raw data, as indicated in the scatterplot on the next page (also shown in our revised Figs. 5b, 6a and 6b). We found that the accuracy of the interpolation relies heavily on how well the raw data match with the Budyko curve, and coupling the residual interpolation can improve the spatial interpolation (see revised Table 2 and Fig. 6).

[Figure]

Comment: "*I'd be happy to give you some support in this process, either regarding interpretation, or to check the interpolation. Just to give some examples of results I find puzzling:*

*- Station ZK is observed to be in the lowest runoff category, the same is the case with all upstream and downstream neighbours. Still it is predicted to be in in the category with more runoff.*"

Reply: This observation could be because of the mistakes in our previous computation routines. In our revised calculations reported in the revision (Fig. 7b), the predicted runoff at station ZK agrees with the runoff at its neighboring stations , e.g., upstream stations GC, XZ, and ZM as well as downstream station BB (seen in revised Fig. 7b).

"*- Almost all stations in the Northern half of the study region are overestimated*"
Reply: In the revised results, this overestimate is not seen.

"*- It is expected that the tiny catchment HWH is underestimated, but it is surprising that it is predicted to have a lower value than all its neighbours.*"
Reply: In the revised results, this underestimate is not seen.

Comment: "*I'm not sure which software you have used for the interpolation. Would it be possible to output the weights for some of the catchments above? That would maybe give some insight in why the results are as they are. Maybe for both the interpolation of raw data and for the interpolation of the residual. It would also be good to see a plot of the residuals.*"

Reply: We carefully checked our software for the interpolation and outputs, including weights at sub-basins. As an example, the table below gives our computed results of the hydro-stochastic interpolation at HWH.

| No. of Basins | Basins | Obs. Runoff (mm) | Weight | Predicted runoff (mm) |
|---|---|---|---|---|

| 12 | HPT | 764.05 | 0.38952 | 297.614 |
|----|-----|--------|---------|---------|
| 23 | QL | 969.78 | 0.31724 | 307.656 |
| 24 | HNZ | 640.04 | 0.21893 | 140.125 |
| 15 | WJB | 293.77 | 0.04260 | 12.515 |
| 11 | ZQ | 117.86 | 0.00661 | 0.779 |
| 3 | SQ | 168.26 | 0.00616 | 1.036 |
| 13 | XX | 367.09 | 0.00577 | 2.119 |
| 21 | ZC | 837.91 | 0.00432 | 3.616 |
| 8 | ZK | 122.58 | 0.00270 | 0.331 |
| 9 | JJJ | 512.69 | 0.00268 | 1.374 |
| 35 | GZ | 342.14 | 0.00094 | 0.322 |
| 40 | HK | 227.05 | 0.00091 | 0.207 |
| 33 | YZ | 235.15 | 0.00090 | 0.211 |
| 36 | DPL | 331.29 | 0.00043 | 0.143 |
| 17 | NLD | 438.87 | 0.00040 | 0.176 |
| 37 | XX2 | 605.85 | 0.00039 | 0.238 |
| 22 | BQY | 693.35 | 0.00019 | 0.128 |
| 6 | XC | 225.24 | 0.00012 | 0.026 |
| 39 | HC | 453.73 | 0.00005 | 0.022 |
| 30 | JZ | 583.17 | 0.00001 | 0.008 |
| 28 | ZT | 437.31 | 0.00001 | 0.006 |
| | Sum | | 1.00000 | 768.650 |

---

## Author Response (AR3)

**Response to the comments of Dr. Skøien**

1. This is a new version of a manuscript describing a coupling of the Budyko approach and hydro-stochastic interpolation. The manuscript has been improved, but there are still a few issues. I also still have some questions regarding the hydro-stochastic interpolation. I am suggesting minor revision here, as there might be good answers to my questions, in that case the editor can accept the manuscript after receiving these. If he is not satisfied with the answers, I'm happy to have another look at a new version.

*Reply: Thanks for the reviewer's work and valuable comments that have helped improve every aspect of our manuscript. In this revision, we have explained your questions regarding the hydro-stochastic interpolation and revised the manuscript according to the reviewer's suggestions (P8L155- P10L193, P18L387- P19L406).*

2. I am happy to see that the new version gives better and more sensible results for the hydro-stochastic interpolation. However, in the answer to the previous review, the authors added the weights for one of the stations. These weights puzzled me, and needs some explanation, by checking closer how the weights could occur.
The weights are for the HWH catchment, which is in the bottom part of the domain. The two closest catchments (BLY and a catchment where I cannot see the name) are not even mentioned in the list of weights. QL and HNZ are quite highly correlated, but still get large weights both of them. This is rather weird, even if hydro-stochastic interpolation is depending on the configuration of the observations and their spatial supports in addition to the distance between them.
Does the software have a correction for negative weights? That could be an explanation, but then it needs to be mentioned. Still it is more common to see the observations "in the shade" getting negative weights. By that I mean that I would expect a relatively large weight for BLY and the unnamed catchments, and negative weights for QL and/or HNZ. If negative weights is the cause, it is necessary to check how large they are before rescaling. Just deleting them and rescaling the rest might not be the best solution, it might be better to pick fewer neighbours for the interpolation in the first place.

*Reply: To answer these questions, we added the calculated weights of HWH catchment and the two closer catchments (BLY and XHD – the unnamed before but we added the name of XHD now in the revision) that were not in the previous list of weights. We checked our calculations and found that:*
*(1) The two closest catchments are HPT and BLY, not BLY and XHD.*
*(2) Yes, there are corrections for negative weights in our software. As the reviewer pointed out, it is common to see the observations getting negative weights. According to the reviewer's suggestions, fewer neighbors of HWH (e.g., the eight stations in Table 1) were selected in our interpolation procedure. There were still negative weights for BLY, XHD, and MS catchments. The negative weight is small at BLY, and a little large at XHD and MS.*
*(3) In our previous interpolation, we deleted the catchments with the negative weights*

*and rescaled the remaining weights. We noticed that deleting the stations with negative weights may not obtain the best solution. In our study, in order to guarantee that the interpolation of $Z^{**}(u)$ using the rescaled weights does not reduce the estimation accuracy of the runoff $Z^*(u)$, we rescaled the weights using the nonlinear programming (a posteriori correction): $\min \left(Z^*(u) - Z^{**}(u)\right)^2$, s.t. $\sum_k^r \lambda_j{}' = 1$ and $0 < \lambda_j{}' < 1, j = k, \dots r.$*

*In terms of the eight stations in Table 1, the weighted runoff using the rescaled positive weights for HPT, ZC, QL, HNZ, and JJJ catchments is much closer to the original weighted runoff with negative.*

So, our rescaled weights don't affect the conclusions that our coupled Budyko and hydro-stochastic interpolation method is better than the Budyko and the hydro-stochastic analysis alone.

*We didn't describe the rescaled method to avoid distraction or confusion and also to ease the reading for readers.*

Table 1 Covariance and weights of HWH

| No. | Sub-basins | Cov(A, B) | Runoff (mm) | weights | weighted R (mm) | Rescaled weights | weighted R using rescaled weights (mm) |
|---|---|---|---|---|---|---|---|
| 1 | HPT | 218905 | 764 | 1.586 | 1212 | 0.789 | 603 |
| 2 | BLY | 210664 | 868 | -0.013 | -12 | 0 | 0 |
| 3 | ZC | 162552 | 838 | 0.064 | 53 | 0.035 | 30 |
| 4 | XHD | 114354 | 740 | -0.760 | -562 | 0 | 0 |
| 5 | QL | 71889 | 970 | 0.164 | 159 | 0.125 | 121 |
| 6 | MS | 32072 | 672 | -0.202 | -136 | 0 | 0 |
| 7 | HNZ | 24772 | 640 | 0.088 | 57 | 0.039 | 25 |
| 8 | JJJ | 14855 | 513 | 0.027 | 14 | 0.013 | 6 |
| sum | | | | | 784 | 1.00 | 784 |

Minor points
1. Section 2.2 is still too long – 3 pages is not necessary for methodologies that have been published before.

*Reply: We shortened section 2.2 but kept the main points of that interpolation method.*

2. In P16, it is mentioned that the area is divided into a 40x50 grid. I find this relatively course resolution for many of the smaller sub-basins, which are likely to get rather few grid points for the calculation of the Ghosh-distance (not geostatistical distance), if I understand correct.

*Reply: Here the geostatistical distance between sub-basin A and B is also called Ghosh*

*distance. It was calculated by averaging the distance between pair of gridded points in two different sub-basins.*

*We compared effects of the grid resolution on the geostatistical distance and the derived function of the covariogram. For example, one test was to double the resolution of the grid from 40×50 grid to 80×100 grid. The geostatistical distance changed from 94.54 km in the coarser grid to 89.13 km in the finer grid between the two sub-basins TJH and XX2 (using them as examples). Our additional tests have shown that this change (doubling the resolution) causes little change in the derived/fitted function for the covariogram in the HRB (the curve fit in Fig. 4). Thus, this resolution has a trivial effect on our interpolation results within the practical limit.*

3. Section 5 is very much a repetition of the summary from 4.4. Some more discussion would instead be useful, such as how does the results in this paper compare to similar studies before.

*Reply: We rewrote this section in the revision according to this suggestion.*

Edits

There are still many grammatical errors – often related to articles and plural/singular. Already in the start of the abstract, it should either read "A Hydro-stochastic interpolation method…" or maybe better: "Hydro-stochastic interpolation methodS based on traditional block kriging HAVE often … A caveat in such methodS ARE that…" There are several more examples in the manuscript, such as P5L84 – such AN approach. P6 L114-119 "THE semi-empirical …" "into A deterministic trend" "calculated as THE difference …"

*Reply: Yes, we noticed issues with use of articles, and have asked for help to correct them. We hope the revision reads better.*

1. P4L63 Blöschl

*Reply: We have changed it.*

2. P4L65-66 I find it unclear what this sentence really says. It should anyway be constrains.

*Reply: This sentence has been revised as "The river network constrains the water paths from upstream to downstream in a basin."*

3. P4L73 What is an "integrated course"? Rephrase

*Reply: It has been changed to "integrated process."*

4. P5L80 I think there are more important reasons for the deviations than the influence FROM heterogeneous rainfall.

*Reply: This sentence has been revised as "The observed patterns of runoff reveal systematic deviations from the homogeneity assumption, however, because of the influences from the heterogeneous climate and underlying surface factors."*

5. P5L86 "method FOR describing complex runoff patterns suffers FROM AN inevitable …"
*Reply: According to the reviewer's suggestion, it has been modified as "... the deterministic method for describing complex runoff patterns suffers from an inevitable loss of information."*

6. P5L90-93 KED is not recent – maybe rather something like "A method that combines both deterministic patterns and stochastic variability is kriging … " And then "It takes deterministic patterns of spatial variables into account and incorporates these as a local trend, a smoothly varying secondary variable, instead of a function of spatial coordinates."
*Reply: These sentences from P5L90-93 are revised to be "A method that combines both deterministic patterns and stochastic variability is the kriging with an external drift (KED) (Goovaerts, 1997; Li and Heap, 2008; Laaha et al., 2013). It takes the deterministic patterns of spatial variables into account and incorporates them as a local trend of a smoothly varying secondary variable, instead of a function of the spatial coordinates."*

7. P6L117-118 comma after residuals, remove "both of"
*Reply: It has been revised.*

8. P13L266 "millions of tons of water" – this is not very precise, and actually not a particularly large number for a region of this size. As an example, 10 million m3/year equals 0.3 $m^3$/second.
*Reply: This sentence has been changed to "About 18 billion $m^3$ of water was consumed in 1998 to meet the basin's domestic and agriculture needs."*

**The list of all relevant changes in the revised manuscript**

1. Some grammatical errors in the manuscript, related to the use of articles and plural/singular, have been corrected;
2. P4L65: It has been changed to "Blöschl";
3. P4L67: It has been changed to "The river network constrains the water paths from upstream to downstream in a basin";
4. P4L75: It has been changed to "…integrated process…";
5. P5L80-82: This sentence has been revised as "The observed patterns of runoff reveal systematic deviations from the homogeneity assumption, however, because of the influences from the heterogeneous climate and underlying surface factors";
6. P5L88-89: Some words have been changed in this sentence;
7. P5L93-97: These sentences have been revised according to the reviewer's suggestion;
8. P6L120-121: "both of" in the original sentence has been deleted and a comma has been added;
9. P8L155- P10L193: This section has been shortened and the main points of the method has been kept;
10. P12L239-240: The sentence has been revised;
11. P18L387- P19L406: The section of discussions and conclusions has been rewritten;
12. P21L447-449: A reference has been added;
13. P26: Some articles have been added in the captions of the figures;
14. P27, P29: Some articles have been added in the captions of the tables;
15. The name of "XHD" has been added in Figure 1, 2, 3 and 7.

**Hydro-Stochastic Interpolation Coupling with Budyko Approach for Prediction of Mean Annual Runoff**

Ning Qiu[a,b], Xi Chen[d,a,b]*, Qi Hu[c], Jintao Liu[a,b], Richao Huang[a,b], Man Gao[a,b]

[a] *State Key Laboratory of Hydrology-Water Resources and Hydraulic Engineering Hohai University, Nanjing 210098, China*
[b] *College of Hydrology and Water Resources, Hohai University, Nanjing 210098, China*
[c] *School of Natural Resources, University of Nebraska-Lincoln, Lincoln NE 68583, U.S.*
[d] *Institute of Surface-Earth System Science, Tianjin University, Tianjin China*

*Corresponding author          E-mail: xichen@hhu.edu.cn*

**Abstract**

The hydro-stochastic interpolation method based on the traditional block-kriging has often been used to predict mean annual runoff in river basins. A caveat in such method is that the statistic technique provides little physical insight on relationships between the runoff and its external forcing, such as the climate and land-cover. In this study, the spatial runoff is decomposed into a deterministic trend and deviations from it caused by stochastic fluctuations. The former is described by the Budyko method (Fu's equation) and the latter by stochastic interpolation. This coupled method is applied to spatially interpolate runoff in the Huaihe River Basin of China. Results show that the coupled method significantly improves the prediction accuracy of the mean annual runoff. The error of the predicted runoff by the coupled method is much smaller than that from the Budyko method and the hydro-stochastic interpolation method alone. The determination coefficient for cross-validation, $R^2_{cv}$, from the coupled method is 0.87, larger than 0.81 from the Budyko method and 0.71 from the hydro-stochastic interpolation. Further comparisons indicate that the coupled method also has reduced the error in overestimating low runoff and underestimating high runoff suffered by the other two methods. These results support that the coupled method offers an effective and more accurate way to predict the mean annual runoff in river basins.

**Keywords:** Coupled Budyko and hydro-stochastic interpolation method; mean annual runoff; prediction accuracy; Huaihe River Basin

**1. Introduction**

The runoff observed at the outlet of a basin is a crucial element for investigating the hydrological cycle of the basin. Because runoff is influenced by both deterministic and stochastic processes, estimating the spatial patterns of runoff and associated distribution of water resources in ungauged basins has been one of the key problems in hydrology (Sivapalan et al., 2003), and a thorny issue in water management and planning (Imbach, 2010; Greenwood et al., 2011).

In estimating and predicting runoff and regional water resources availability, we have often used regional or global runoff mapping and geostatistical interpolation methods. In these methods, the value of a regional variable at a given location is often estimated as the weighted average of observed values at neighboring locations. This interpolation of runoff, which is assumed as an auto-correlated generalized stochastic field (Jones, 2009), uses secondary information from more than one variable (Li and Heap, 2008). Spatial autocorrelations of the runoff values are measured by the covariance or semi-variance between the runoffs at pairs of locations as a function of their Euclidian distance (such as in the ordinary kriging). The values obtained by the interpolation methods are the best linear unbiased estimate in the sense that the expected bias is zero and the mean squared error is minimized (Skøien et al., 2006). The ordinary kriging (OK) estimates the local mean as a constant; corresponding residuals are considered as random. Because the spatial mean could also be used as a trend or nonstationary variation in space, OK has been developed into various geostatistical interpolation methods, such as kriging with a trend by incorporating local trend within a confined neighborhood as a smoothly varying function of the coordinates. Block kriging (BK) is another extension of OK for estimating a block value instead of a point value by replacing the point-to-point covariance with point-to-block covariance (Wackernagel,

1995).

Unlike precipitation or evaporation which we often interpolate to find its values at specific locations, runoff is an integrated spatially continuous process in river basins (Lenton and RodriguezIturbe, 1977; Creutin and Obled, 1982; Tabios and Salas, 1985;

Dingman et al., 1988; Barancourt et al., 1992; Blöschl, 2005). Streamflows are naturally organized in basins (Dooge, 1986; Sivapalan, 2005), e.g., rivers flow through sub-basins.

The river network constrains the water paths from upstream to downstream in a basin.

The hierarchically organized river network requires that the sum of the interpolated discharge from sub-basins equals to the observed runoff at the outlet of the entire basin.

Previous studies have indicated that runoff interpolation may overestimate the actual runoff without adequate information of the spatial variation of the runoff (Arnell, 1995), e.g., neglecting the river network in connecting sub-basins or processing basin runoff at collective points in space (Villeneuve et al, 1979; Hisdal and Tveito, 1993). In nested basins, Gottschalk (1993a and b) developed a hydro-stochastic method to interpolate runoff. It uses the concept that runoff is an integrated process in the hierarchical structure of river network. Distance between a pair of basins is measured by geostatistical distance instead of the Euclidian distance. The covariogram among points in the conventional spatial interpolation is replaced by the covariogram between basins. In this concept, runoff is assumed spatially homogeneous in basins, i.e., the expected value of the runoff is constant in space (Sauquet, 2006). The observed patterns of runoff reveal systematic deviations from the homogeneity assumption, however, because of the influences from the heterogeneous climate and underlying surface factors.

An alternate method is to describe the hydrological variables of interest in deterministic forms of functions, curves or distributions, and construct conceptual and mathematical models to predict hydro-climate variability (Wagener et al, 2007). Qiao (1982), Arnell (1992), and Gao et al. (2017) have used such an approach and derived empirical relationships between runoff and its controlling factors of the climate, land- cover, and topography in various basins. However, the deterministic method for describing complex runoff patterns suffers from an inevitable loss of information (Wagener et al, 2007) because of existence of uncertainty in many hydrological processes and especially in observations. Thus, hydrological variables also contain the information of stochastic nature and should be treated as outcomes from deterministic and stochastic processes. A method that combines both deterministic patterns and stochastic variability is the kriging with an external drift (KED) (Goovaerts, 1997; Li and Heap, 2008; Laaha et al., 2013). It takes the deterministic patterns of spatial variables into account and incorporates them as a local trend of a smoothly varying secondary variable, instead of a function of the spatial coordinates.

The inclusion of deterministic terms in the geostatistical methods has been shown to increase the interpolation accuracy of basin variables, such as mean annual runoff (Sauquet, 2006), stream temperature (Laaha et al., 2013), and groundwater table (Holman et al., 2009). Those deterministic terms are often described by empirical formulae linking spatial features, e.g., variability of the mean annual runoff in elevation (Sauquet, 2006), and relationship between the mean annual stream temperature and the altitude of gauges (Laaha et al., 2013). As a semi-empirical approach to model the deterministic process of the runoff, the Budyko framework has been popularly used to analyze the relationship between mean annual runoff and the climatic factors, e.g., aridity index (Milly, 1994; Koster and Suarez, 1999; Zhang et al., 2001; Donohue et al.,

2007; Li et al., 2013; Greve et al., 2014). Many efforts have been devoted to improving the Budyko method by, for example, including the effects of other external forcing factors, such as land-cover (Donohue et al., 2007; Li et al., 2013; Han et al., 2011; Yang et al., 2007), soil properties (Porporato et al., 2004; Donohue et al., 2012), topography (Shao et al., 2012; Xu et al., 2013; Gao et al., 2017), hydro-climatic variations of seasonality (Milly, 1994; Gentine et al., 2012; Berghuijs et al., 2014), and groundwater (Istanbulluoglu et al., 2012). However, it has been found that the use of the deterministic equation in the Budyko method alone still comes with large errors in the prediction of runoff in many basins (e.g., Potter and Zhang, 2009; Jiang et al., 2015).

The aim of this study is to combine the stochastic interpolation with the semi- empirical Budyko method to further improve the spatial interpolation/prediction of the mean annual runoff in the Huaihe River Basin (HRB), China. In this study, the spatial runoff from sub-basins in the HRB is separated into a deterministic trend and its residuals, which are estimated by the Budyko method and the interpolation method, respectively.

The residuals are calculated as the difference between the observed and the estimated runoff from the Budyko method, and are used in the stochastic interpolation as described in Gottschalk (1993a, 1993b, and 2000). After that, the runoff of any sub-basin is predicted as the sum of the interpolated residuals and the Budyko estimated value. The improved method is tested in the HRB. In addition, the leave-one-out cross-validation approach is applied to evaluate and compare the performances 
[revised manuscript text omitted]

Budyko and the hydro-stochastic interpolation method, we compare and contrast its performance with the Budyko and the hydro-stochastic interpolation method alone.

Their performances are evaluated by the following metrics (Laaha and Bloschl, 2006):

$$MAE = \frac{1}{n}\sum_{j=1}^{n}[R(x_i) - R^*(x_i)]$$                    (9)

$$MSE = \frac{1}{n}\sum_{j=1}^{n}[R(x_i) - R^*(x_i)]^2$$                    (10)

$$RMSE = \sqrt{\frac{1}{n}\sum_{j=1}^{n}[R(x_i) - R^*(x_i)]^2}$$                    (11)

where, $R^*(x)$ and $R(x)$ are the predicted and the observed runoff, respectively, *MAE*

is the mean absolute error, *MSE* is the mean square error, and *RMSE* is the root-mean- square error. The determination coefficient for cross-validation is

$$R_{cv}^2 = 1 - \frac{V_{cv}}{V_{NK}}$$                    (12)

where, $V_{cv}$ is the mean square error (*MSE*), and $V_{NK}$ is the spatial variance ($V_{NK} =$

$\frac{\sum_{j=1}^{n}[R(x_i) - \bar{R}]^2}{n-1}$, in which $\bar{R}$ is the mean $R(x)$) of the runoff over all the tested sub-basins.

In addition to these evaluation metrics, the prediction result is evaluated by regression analysis of the observation vs. the prediction.

**3. Study catchment and data**

The Huaihe River Basin (HRB) – the sixth largest river basin in China, is used in evaluation of our coupled model and in its comparison to the other two methods. The HRB has a strong precipitation gradient from the humid climate in the east and the semi-humid in the west (Hu, 2008). It is one of the major agricultural areas in China with the highest human population density in the country. About 18 billion $m^3$ of water was consumed in 1998 to meet the basin's domestic and agriculture needs. Water resources per capita and per unit area is less than one-fifth of the national average. Moreover, more than 50% of the water resources is exploited, much higher than the recommended 30% for inland river basins (Yan et al., 2011). Moreover, the concentrated annual precipitation in a few very rainy months makes the region highly vulnerable to severe floods or droughts (Zhang et al., 2015). Thus, having the knowledge of the spatial distribution of the runoff is vital for water resources planning and management in the region.

Our study area is in the upstream of the Bengbu Sluice in the HRB and is 121,000 $km^2$ (Fig. 1). The river network in the area is derived from data packages of the National Fundamental Geographic Information System, developed by the National Geomatics Center of China. The HRB is divided into 40 sub-basins, according to available hydrological stations with records from 1961-2000 (Fig. 2). The sub-basins vary in their size from the smallest of 17.9 $km^2$ to the largest of 30630 $km^2$. Among the 40 sub-basins, 27 are independent sub-basins and 13 are nested sub-basins.

Annual precipitation data used in this study are from 1961-2000 and are obtained from a monthly mean climatological dataset at 0.5-degree spatial resolution. The dataset was developed at China Meteorological Administration, and is accessible at:

http://data.cma.cn/data/detail/dataCode/SURF_CLI_CHN_PRE_MON_GRID_0.5.htm l. The dataset was derived from the observations at 2472 stations in China, using the

Thin Plate Spline (TPS) interpolation method and the ANUSPLIN software. Pan evaporation data at 21 meteorological stations in the HRB are used to interpolate $E_0$ by the ordinary kriging method and the ArcGIS. The interpolated $E_0$ are used to derive the annual potential evapotranspiration in the sub-basins. The statistical features of the mean annual precipitation ($P$), $E_0$, and the runoff depth ($R$) from 1961-2000 are summarized in Table 1. They show that $P$ varied between 638-1629 mm, annual temperature was between 11°-16°C, and the mean annual $E_0$ between 900-1200 mm. The sub-basins in the north, e.g., ZM, ZQ, XY, and ZK in Fig. 2, are relatively dry with the dryness index (E$_0$/P) above 1.3. The sub-basins in the south, e.g., MS, HBT, and HC, are wetter with dryness index below 0.8. The average mean annual $R$ is about 400 mm, fluctuating from

90 mm in the north to 1000 mm in the south. The temporal and spatial variations in the runoff are relatively small in the south and large in the north.

**4 Results**

**4.1 Prediction of runoff by the Budyko method**

Actual evapotranspiration $E$ is estimated using long-term mean annual water balance ($E=P-R$) from 1961–2000 at the 40 sub-basins, and the results are shown in Table

1. Also shown in Table 1 are the calculated $\omega$ values for the sub-basins. They vary from

1.43 in the sub-basin HWH to 3.16 in JJJ. The average $\omega$ is 2.32 for the 40 sub-basins.

The comparison $E/P$ vs. $E_0/P$ is shown in Fig. 3. The best fit (curve) for $E/P$ vs. $E_0/P$, or

$R$ vs. $E_0/P$, is also shown in Fig, 3; it gives an alternative for average $\omega$ of the sub-basins.

The fitted value of $\omega$ for the 40 sub-basins determined from this process is 2.213, very close to that calculated directly from the 40 individual sub-basins.

Using $\omega=2.213$ in the HRB, Fu's equation in Eq. (2) can be written as

$$R = P \cdot \left(1 + \left(\frac{E_0}{P}\right)^{2.213}\right)^{\frac{1}{2.213}} - E_0. \tag{13}$$

Eq. (13) and Fig. 3 clearly show the deterministic trend of the runoff in the HRB.

According to the water limit criterion, $E = P$, and the energy limit criterion, $E = E_0$, in

Fig. 3a, the smaller the index $\frac{E_0}{P}$ is the smaller the $\frac{E}{P}$ will be (Fig. 3a) or the larger the runoff will be (Fig. 3b) from the sub-basins in the HRB. In Figs. 3b and 3c, the lower $R$

in the northern sub-basins indicates drier conditions ($E_0/P>1.4$), whereas the higher $R$ in the southern sub-basins assures wetter conditions ($E_0/P<0.8$).

Using $P$ and $E_0$ given in Table 1 for the 40 sub-basins, we predict the runoff $R$ by

Eq. (13), the Budyko method, and the deviations of their predictions from the observation. The results are summarized in Tables 1 and 2. The *MAE* of predicted $R$ is

94 mm, and *RMSE* is 112 mm. The largest absolute error is in the sub-basin HWH (328

mm), and the smallest in XX (24 mm). The largest relative error is 81.6% of the observed runoff in the sub-basin XZ, and the smallest is 5.0% of the observed runoff in XHD.

They represent absolute errors of 91 and 37 mm in those two sub-basins, respectively.

**4.2 Runoff by the hydro-stochastic interpolation method**

For comparison, the observed runoff is used in the hydro-stochastic interpolation following the procedure detailed in section 2.2. In order to obtain the distance $d$

between pairs of the sub-basins, the study area is divided into 40 row×50 column. The geostatistical distance between any two sub-basins, A and B, is calculated by averaging the distances between all pairs of grid points in A and B (all the possible pairs of the sub- basins are 40×41/2 for the 40 sub-basins in this study). According to the estimated distance for the pairs of sub-basins and the observed runoff at the 40 sub-basins (Table

1), the empirical covariance $Cov_e(d)$ is estimated for each pair of the sub-basins. From the plots of the mean $Cov_e(d)$ of all the independent sub-basin pairs vs. the corresponding distance $d$ with an interval of 20 km, we fit the function of empirical covariogram shown in Fig. 4. The fitting theoretical covariance function $Cov_p(d)$ to the empirical covariogram is

$$Cov_p(d) = 6 \times 10^5 \exp(-d/28.62). \tag{14}$$

This function is used to calculate the average theoretical covariance $Cov(A,B)$ in Eq. (7).

Finally, the weight matrices are determined using our programs in MatLab.

The interpolated runoff depth ($R$) over the 40 sub-basins along with the deviations from the observation are shown in Table 1. The *MAE* and *RMSE* of $R$ are 103 and 140

mm, respectively. The largest absolute and relative error is in the sub-basin JZ (401 mm and 68.8%), and the smallest is in DPL (1 mm and 0.3%) (Table 2). These results indicate that the errors from this interpolation method are in general larger than those from the

Budyko method, suggesting that the observed runoff is more influenced by the deterministic trend in the basin.

**4.3 Hydro-stochastic interpolation with Fu's equation (our coupled method)**

We use Fu's equation, Eq. (2), to evaluate the deterministic trend or the external drift function, $R_d{}^*(x)$, and deviation of the trend from the observation, $R_s{}^*(x)$, assuming a spatially auto-correlated process. The $R_s{}^*(x)$ is then used in the stochastic interpolation.

The empirical residual covariogram of $R_s{}^*(x)$ for each pair of sub-basins vs. sub- basin distance is shown in Fig. 5. From the result in Fig. 5a, we obtain the exponential function for $Cov_p(d)$

$$Cov_p(d) = 13030 \exp(-d/23.9).$$    (15)

From (15), the weight matrices of runoff deviation are determined by Eq. (4) using our program in MatLab. They are then used to predict the runoff deviation. The scatterplot of the predicted residuals vs. the observed residuals shown in Fig. 5b delineates a positive correlation between the predicted and the observed residuals. However, the large scatter indicates limited performance by the residual model alone. Because this interpolation scheme represents the spatial runoff deviation, the sum of the interpolated runoff deviation and the simulated runoff by Fu's equation is the total interpolated runoff in the sub-basins.

The predicted runoff using this procedure is given in Table 1, with the *MAE* at 71

mm and *RMSE* at 93 mm over the 40 sub-basins. The largest absolute error is in the sub- basin QL (220 mm), and the smallest in ZM (4 mm) (Table 2). The largest relative error is 47.2% of the observed runoff in XZ, and the smallest is 1% of the observed runoff in

BLY. They represent the absolute error of 52 and 8 mm, respectively.

**4.4 Comparisons of the predicted runoff by the three methods**

Comparing the results in Table 2, we find that our coupled method of the deterministic and stochastic processes substantially reduces the runoff prediction error in the HRB. The *MAE* and *RMSE* of the runoff from our coupled method are much smaller than those from the Budyko or the hydro-stochastic interpolation method. In cross-validation (Table 2), our coupled method has $R_{cv}^2$=0.87, which is larger than 0.81

and 0.71 from the Budyko method and the hydro-stochastic interpolation, respectively.

The errors in runoff at the sub-basins are significantly reduced as well. The error in the sub-basin HWH is 216 mm from the coupled method, compared to 328 mm from the

Budyko method and 300 mm from the hydro-stochastic interpolation. The error in JZ is

120 mm from the coupled method, smaller than 179 mm from the Budyko method and

401 mm from the hydro-stochastic interpolation.

Our correlation analysis between the predicted and the observed *R* is shown in Fig.

6. The predicted runoff from our coupled method shows higher correlation with the observed ($R^2$=0.87), in comparison to the Budyko method ($R^2$=0.82) and the hydro- stochastic interpolation ($R^2$=0.79). Our analysis indicates that the latter two methods overestimate low runoff and underestimate high runoff, as indicated by large departures from the 1:1 line in Fig. 6. Similarly, large deviations of the runoff predicted by the hydro-stochastic interpolation have also been reported by Sauquet et al. (2000), Laaha and Bloschl (2006), and Yan et al. (2011).

The spatial distributions of the runoff in the HRB calculated from the three methods are shown in Fig. 7. They again show significant differences. Compared to the result from our coupled method (Fig. 7c), the Budyko method overestimates the runoff in most of the northern sub-basins (Fig. 7a), where the climate is relatively dry and runoff is small (ranging from 140-280 mm). The hydro-stochastic interpolation method underestimates the runoff in some southern sub-basins (Fig. 7b), where the wet climate has fostered extremely high runoff (800~1100mm), such as in the sub-basins HWH, BLY, and ZC (Table 1). The results from our coupled method are closest to the observed distribution of the runoff among the three methods (Fig. 7d). Compared to the errors in the predicted runoff by the Budyko method and the hydro-stochastic interpolation (Fig.

7 and Table 1), our coupled method reduces the error in 70% of all the sub-basins (28 of the 40 sub-basins).

**5. Discussions and conclusions**

In this study, we use the Budyko's deterministic method to describe the mean annual runoff, which is an integrated spatially continuous process and determined by both the hydro-climatic elements and the hierarchical river network. A deviation from the Budyko estimated runoff is used by the stochastic interpolation that assumes spatially auto- correlated error. The deterministic aspects of the runoff described in Budyko method are reflected in the trends at locations (sub-basins), and deviations from the trends caused by the stochastic processes are described by the weights as a function of the autocorrelation and distance. Information from both the Budyko method and the stochastic interpolation are integrated in our coupled method to predict the runoff.

Different from the universal kriging method, in which the trend is represented as a linear function of coordinate variables and determined solely through spatial data calibration (i.e., semi-variogram analysis), the Budyko method couples water and energy balance and could directly predict streamflow in ungauged basins. This physically based method relies on using the spatial trend of runoff and, in our study, it yields the deterministic coefficient of cross-validation, $R_{cv}^2$, to be 0.81, better than that from the hydro-stochastic interpolation method.

Incorporating secondary information into the geostatistical methods improves the estimate of a predictive variable, e.g., the estimate of groundwater level by incorporating topography into the collocated co-kriging (Boezio et al., 2006), or the estimate of mean annual stream temperature by incorporating a nonlinear relationship between the mean annual stream temperature and altitude of the stream gauge into the Top-Kriging (Laaha et al., 2013). By incorporating such secondary information and the relationship between the mean runoff and the climate conditions (the aridity index) in the Budyko method through coupling with the hydro-stochastic interpolation, we develop our new coupled

Budyko-hydro-stochastic interpolation method. It can substantially improve the prediction of streamflow in ungauged basins. This improvement is shown by the higher

$R_{cv}^2$ of 0.87 in the HRB, compared to 0.81 and 0.71 by the Budyko and the hydro- stochastic interpolation method, respectively. Moreover, for high and low runoffs in the sub-basins of the HRB our coupled method gives more accurate predictions.

While substantial progress has been made by our coupled method, its results show rooms for improvement to further increase the accuracy of runoff prediction. For example, runoff prediction errors remain large from our coupled method in some sub- basins in the HRB. In the sub-basins MS, QL, HWH, and HNZ, the absolute error of predicted runoff is larger than 150mm and the relative error of predicted runoff is larger than 20% of the observed runoff. In the sub-basins BGS and XZ, the relative error of the predicted runoff is larger than 40% of the observed runoff. These errors are largely attributable to large prediction errors intrinsic to the Budyko method (e.g., MS, QL,

HWH, and XZ in Table 1). Possible causes to the errors could be from additional external factors influencing the runoff, such as land-cover, soil properties, hydro-climatic variations, and the groundwater. Including some or all these effects to improve the

Budyko method or incorporating these effects as secondary information (e.g., multicollocated co-kriging) in our coupled model would help aid our understanding of the deterministic processes and increase the runoff prediction accuracy.

**Acknowledgement**

We thank the editor Dr. Erwin Zehe, the reviewers Drs. M. Mälicke and J.O. Skøien for their valuable comments and suggestions that helped improve this manuscript substantially. The research was supported by the National Natural Science Foundation of China (No. 51190091 and 41571130071). Qi Hu's contribution was supported by

USDA Cooperative Project NEB-38-088.

[revised manuscript text omitted]

   fit function). (c) The sub-basins in the north and south of the study basin. Note:

   in (b) and (c), blue color indicates wetter climate in the south and yellow color

   indicates drier climate in the north.

Figure 4: Empirical covariogram ($Cov_e(d)$) from the sub-basin runoff data and

   theoretical covariogram by fitted covariance function $Cov_p(d)$ of the study area.

Figure 5: (a) Empirical covariogram ($Cov_e(d)$) from the residual $R_s(x)$ and theoretical

   covariogram by fitted covariance function $Cov_p(d)$ of the study area. (b) The

   scatterplot of the predicted vs. the observed residuals.

Figure 6: Cross validation of the predicted runoff vs. the observation by (a) Budyko

   method, (b) hydro-stochastic interpolation, (c) our coupled method, and (d) the

   scatterplot of the predicted vs. the observed residuals for (c). The dashed-line is

   1:1.

Figure 7: Spatial distribution of the mean annul runoff estimated from (a) Budyko

   method, (b) hydro-stochastic interpolation, (c) our coupled method, and (d) the

   observation.

     Table 1: Summary of hydro-meteorological data and predicted runoff of the sub-basins in the HRB.

[revised manuscript text omitted]